# Taxonomical Insights on Parameter Space Generalization in Over-parameterized Models

## Abstract

In this study we attack the conundrum of success of over-parameterized models from understanding the complex relationship between parameter space and output space.

We classify key parameter sets related to generalization and training in parametric basis expansion machine learning models. Methods ranging from Linear regression, Extreme learning machines to Neural networks fall in this category. We also classify these parametric models into identifiable and non-identifiable models according to the mapping from parameter space to function space. Such a classification of models is already present in literature but usually studied in Bayesian ML and statistics. We focus on identifiable models in this article.

We later classify generalization into strict and weak generalization according to learning in parameter space for fixed basis regression models which fall into the category of model-identifiability. Strict generalization is when true parameters (or their un-identifiable counterparts) of the ground truth are learned, while weak generalization is when they are not learned but we still achieve local generalization. We showcase the conditions needed for strict generalization in fixed basis regression settings, trained using pseudo-inverse methods. We showcase that strict generalization cannot be achieved in over-parameterized regimes trained through pseudo-inverse method, but approaching the strict generalization using gradient descent can be completely dependent on our initialization and randomness. Thus supporting the classical idea that over-parameterization is bad, but emphasizing that it applies to strict generalization case. However, weak generalization can always be achieved in over-parameterized regimes under certain cases. Thus we study the complex relationship between generalization in output space and the parameter space to understand the conundrum of success of over-parameterized models and try to weave a coherent and consistent picture of the same.

Later we study generalization performance under label noise, for the distinct scenarios identified in this article. We include insights into the theory of deep learning and quantum machine learning.

Our work serves as refinement of the idea of generalization as well as it provides insights through proof and sometimes demonstrations. Our focus in this article is purely taxonomical and conceptual rather than driven by introduction of new metrics.

## 1 Introduction

The recent success of deep learning has renewed interest in the generalization behavior of over-parameterized models, a phenomenon that appears counterintuitive under classical learning theory. Traditional theory suggests that models with more parameters than data points should overfit, yet in practice, highly over-parameterized models often achieve low test error (Bishop and Nasrabadi, 2006; Geman et al., 1992). Understanding the mechanisms behind this behavior remains a central challenge in modern machine learning.

Several phenomena have been proposed to explain generalization in over-parameterized regimes. *Double Descent* describes a non-monotonic test error curve, where increasing model complexity initially worsens

generalization but subsequently improves it (Nakkiran, 2019; Nakkiran et al., 2021; Belkin et al., 2019). *Benign overfitting* occurs when models perfectly fit training data yet still generalize well (Bartlett et al., 2020). Other observations include *implicit regularization* in gradient-based optimization (Neyshabur, 2017; Gunasekar et al., 2018), *grokking*, where generalization emerges after prolonged training (Power et al., 2022; de Mello Koch and Ghosh, 2025), and connections to the *Neural Network Gaussian Process (NNGP)* correspondence (Neal, 2012; Lee et al., 2017). While these phenomena provide partial insights, they do not fully explain why over-parameterized models sometimes fail to generalize, particularly in terms of recovering the underlying structure of the target function.

A key limitation of existing explanations is the conflation of low test error with faithful recovery of the target function. In practice, a model can achieve small test error without accurately learning the underlying coefficients or features, especially when evaluated outside the training domain. To formalize this distinction, we introduce two complementary notions of generalization: *strict generalization*, which requires accurate recovery of the target function, and *weak generalization*, which requires only low test error on the training domain. These definitions allow us to distinguish between in-domain approximation and out-of-domain extrapolation, and to clarify the success and limitations of over-parameterized models.

To analyze these phenomena rigorously, we focus on a fixed basis regression model, a class encompassing linear regression, random feature models, and extreme learning machines. These models also fall into the category of identifiable models as they are linear in parameters, i.e. there is injective mapping between parameters and the model function class. This framework allows precise characterization of the mapping between parameter space and function space, and facilitates analytical study of generalization in the over-parameterized regime. For these settings, we define two novel thresholds: the *sampling threshold*, which captures the minimal data requirements for parameter recovery, and the *expressivity threshold*, which captures the minimal model complexity required to represent the target function. These thresholds complement the well-known *interpolation threshold* observed in double descent.

Our analysis yields several insights. First, strict generalization is fundamentally unattainable in over-parameterized fixed basis regression when trained using pseudo-inverse methods, due to the inability of minimum-norm solutions to recover true coefficients. Consequently, out-of-domain extrapolation is also impossible in such settings if trained with pseudo-inverse method. In contrast, weak generalization remains achievable: models can approximate the target function within the training domain even without recovering its true structure. Second, using Bernstein basis function as an example and the Weierstrass Approximation Theorem, we prove that weak generalization is theoretically guaranteed for any closed and bounded continuous function in one dimension in highly over-parameterized regimes trained with pseudo-inverse. In such scenarios parameters or its non-identifiable counterparts are not learned. This result highlights that over-parameterization can sometimes enable in-domain approximation regardless of the training method, while extrapolation requires additional constraints or use of optimization schemes rather than complete solutions like Pseudo-inverse.

By establishing a clear distinction between strict and weak generalization, introducing novel thresholds for fixed basis regression models, and providing rigorous guarantees for in-domain approximation, our work offers a refined theoretical perspective on over-parameterized learning. Our primary motive in this article is of classification of the modes of generalization and its behavior. These insights can clarify the success and limitations of deep learning and extend naturally to related frameworks, including quantum machine learning.

## 1.1 Relevant Phenomena and Related Works

Over-parameterized models exhibit several phenomena that partially explain their generalization behavior, yet key gaps remain.

**Double Descent:** Contrary to classical bias–variance intuition, test error can initially increase and then decrease as model complexity grows (Nakkiran, 2019; Nakkiran et al., 2021; Belkin et al., 2019). Variants include model-wise, sample-wise, and epoch-wise double descent. Several questions still remain, for example, while double descent illustrates that over-parameterization need not induce catastrophic overfitting (Bartlett et al., 2020), it does not guarantee accurate recovery of the target function. Similar insights have been ex-

plored in quantum machine learning, emphasizing feature encoding and generalization (Peters and Schuld, 2023). These works largely address strict generalization, with limited consideration of in-domain approximation or weak generalization. Other studies have examined the role of feature matrix condition numbers in double descent (Poggio et al., 2019), but their scope is restricted to random matrices and does not clarify how model basis choice or identifiability conditions affect approximation quality.

**Benign Overfitting:** Certain over-parameterized models perfectly fit training data yet generalize well (Bartlett et al., 2020). This behavior depends on model structure, data distribution, and optimization method. Importantly, low test error does not imply faithful function reconstruction, highlighting the distinction between weak and strict generalization.

**Implicit Regularization:** Optimization algorithms such as gradient descent bias solutions toward minimum-norm or low-rank parameter configurations (Neyshabur, 2017; Gunasekar et al., 2018; Soudry et al., 2018). Such implicit biases can facilitate generalization without explicit regularization, but their effect is limited to particular optimization schemes and initialization regimes.

**Grokking:** Delayed generalization emerges after prolonged training despite early overfitting (Power et al., 2022; de Mello Koch and Ghosh, 2025). This phenomenon emphasizes temporal dynamics of learning and suggests that standard static analyses may be insufficient to fully characterize generalization.

**NNGP and Mean-Field Correspondences:** Infinite-width neural networks converge to Gaussian processes (NNGP) (Neal, 2012; Lee et al., 2017) or admit mean-field PDE descriptions of gradient descent dynamics (Mei et al., 2018). While these frameworks reveal inductive biases of architectures, they often ignore finite-width effects and feature learning dynamics (Jacot et al., 2018; Golikov et al., 2022).

**Singular Learning Theory:** By considering the many-to-one mapping from parameter space to function space, singular learning theory introduces the real log-canonical threshold (RLCT) to quantify effective model complexity (Watanabe, 2024; Wei et al., 2022). Although this theory advances understanding of over-parameterized models, it primarily addresses post-training scenarios and Bayesian generalization, leaving open questions about dynamic learning and in-domain vs out-of-domain approximation.

Other heuristic insights include the Lottery Ticket Hypothesis (Frankle and Carbin, 2018), spectral bias (Rahaman et al., 2019), and ReLU network spline interpretations (Sahs et al., 2022; Balestriero et al., 2018).

## 1.2 Our contribution

We emphasize that

## 1.3 Outline of the Article

The remainder of the manuscript is organized as follows:

- section 2 – **Problem Setup:** We formalize the regression task, define the fixed basis model, and introduce key concepts including strict vs weak generalization and the sampling and expressivity thresholds. We define identifiability of a model and focus our work to identifiable models. We discuss Sampling schemes and their influence on generalization.

- section 3 – **Strict Generalization:** We analyze conditions for strict generalization in over-parameterized regimes and prove its impossibility using pseudo-inverse training. Implications for out-of-domain extrapolation are presented.

- section 4 – **Weak Generalization:** We demonstrate that weak generalization is achievable in-domain, even in over-parameterized settings. Using Bernstein basis functions, we show that any one-dimensional closed and bounded continuous function can be approximated near training points.

- section 5 – **Effect of Noise:** We study the impact of sampling noise and model basis choice on generalization, highlighting the relative importance of basis selection and regularization.

- section 6 – **Parameter Classification:** We classify parameters in the model space according to their generalization properties and clarify the conditions under which strict generalization may be recovered via optimization-based training even in over-parameterized regime.

- section 7 – **Insights for Deep Learning:** We discuss implications of our findings for over-parameterized neural networks, including when weak generalization emerges and the role of architecture and initialization.

- section 8 – **Insights for Quantum ML:** We extend the framework to quantum machine learning models and discuss implications for feature encoding and generalization.

Appendices: Technical details on basis properties, sampling procedures, proofs of corollaries, noise stability, and a real-world example.

## 2 Setup

Consider the problem of approximating a 1-d continuous function on a closed and bounded domain ($[a, b]$) denoted by $g(x)$. We call it as the *target/true function* or *ground truth*.

Let $n_{train}$ training points be sampled from the function, restricted to domain $[a_{train}, b_{train}]$, with some sampling error $\epsilon$, such that the sampled training points are $\tilde{g}(x_i^{train}) = g(x_i^{train}) + \epsilon_i$, where $i = 0, ...., n_{train} - 1$. This gives us the training dataset $(x_i^{train}, \tilde{g}(x_i^{train}))$. The test of the approximation capability of a model is in the domain $[a, b]$. If $[a, b] = [a_{train}, b_{train}]$, then it is considered *in-domain approximation* and if $[a_{train}, b_{train}] \subset [a, b]$ it is considered *out-of-domain approximation* (extrapolation). The test set is given as $(x_i^{test}, g(x_i^{test}))$. Note, we do not consider any sampling error in the test set as we intend to understand the theoretical approximation capabilities of a model.

Before moving further let us recall the difference between interpolation and approximation. While interpolating a function we intend our model to pass through the training points, while in approximation we care more about approximating the underlying function represented by the data. Overfitting happens when the function passes through the training points, but does not approximate the function. The latter part is important. Hence, as you can predict already, overfitting is a nuisance in the approximation tasks and not for interpolation. If there is no sampling noise, a good approximation should also pass through the training points.

### 2.1 Sampling

Let the true function (in the domain $[a, b]$) be

$$g(x) = \sum_{j=0}^{d-1} \phi_t(x; w_j) c_j^{\text{true}}, \tag{1}$$

where $d \in \mathbb{Z}_+$ is the number of "continuous basis functions" ($\phi_t(x, w)$) in linear expansion of the true function. There is no restriction on the value of $d$ and it can be infinite too. The combination of $d$, $\phi_t(x; w_j)$ and $c^{\text{true}}$ generates different continuous functions in the domain of the basis used. If $\phi$ are polynomial bases, $w$ represents the degree of the polynomial or if it is non-polynomial basis like sine or cosine, it is the frequency. More generally, in our article we consider $w$ to be fixed set of hyper-parameters and belonging to the set $\{w_0, ...., w_{d-1}\}$.

Note, that basis expansion type models for regression like polynomial regression, neural networks or kernel methods (implicitly) ideally should fit functions which can be exactly expanded into a certain basis. Hence we consider functions of the type Equation 1. However, most functions in applications cannot be expanded as such. In such scenarios, we perform *surrogate modeling*, i.e., approximate the ground truth by some basis expansion, given by the Equation 1. Hence, in practice, we are indirectly approximating functions of the type Equation 1. For example, approximation of gravitational potential of Earth which is a complicated function, but we approximate it by a Legendre polynomial of certain order according to the type of application the

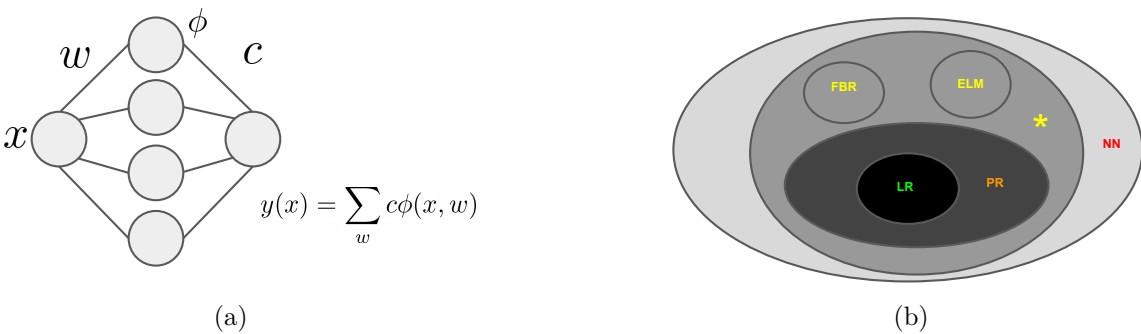

(a)                                                     (b)

Figure 1:   (a) Schematic comparison of fixed basis regression model and feed-forward neural network. The neural networks can be called as adaptive basis regression model. In the latter case we learn the weights (and bias), while in the previous case it is fixed. This understanding can help generalize the results and understanding of this article to deep learning. (b) Diagram showing taxonomy of basis expansion models. Linear regression model is the most restricted case of it. *LR* represents Linear Regression, *PR* represents Polynomial (monomial basis) regression, *FBR* represents Fixed basis regression (the model we use in this article), *ELM* represents Extreme Learning Machine model, **\*** represents the group of all model which can be non-linear in features but linear in learnable weights. When we generalize outside these models we achieve the neural network model.

approximation will be used Hofmann-Wellenhof and Moritz (2006). We discuss such a scenario when the ground truth is not of the type Equation 1 through a real-world example in more detail in Appendix E.

The sampled training point vector can be written as

$$\tilde{\bar{g}}^{train} = \bar{g}^{train} + \bar{E} = \Phi_t^{train}\bar{c}^{true} + \bar{E}, \tag{2}$$

where $\bar{E} \in \mathbb{R}^{n_{train}}$ represents the error vector in sampling, $\Phi_t^{train} \in \mathbb{R}^{n_{train} \times d}$ and $c^{true} \in \mathbb{R}^d$, and $\Phi_t^{train}$ is the true feature matrix defined in training domain. In general in this article we use the symbol $\Phi$, possibly with subscripts and superscripts, to denote a matrix with entries $\Phi_{ij} = \phi(x_i, w_j)$.

The inputs can be sampled uniformly spaced, randomly or with certain sequence, i.e. at nodes of polynomials. We define it in Appendix B.

## 2.2   Model

To approximate the functions consider a model defined as

$$y(x_i) = \sum_{j=0}^{p-1} \phi_m(x_i; w_j)c_j \qquad\qquad \bar{y} = \Phi_m\bar{c}, \tag{3}$$

where $p \in \mathbb{Z}_+$. The model is a linear combination of the basis functions $\phi_m$ generated at points $x$. If $x = x_{train/test}$ then $\Phi_m = \Phi_m^{train/test}$ and $\Phi_m^{train/test} \in \mathbb{R}^{n_{train/test} \times p}$. The coefficient vector is $\bar{c} \in \mathbb{R}^p$. At the training points we consider

$$\bar{y}^{train} = \tilde{\bar{g}}^{train}. \tag{4}$$

Let us try to understand the relation of this model to other well-known methods in machine-learning Figure 1.

We consider $w$ to be fixed (in both the target function and the model) in this article for simplicity. However, the coefficients $c$ are learnable. Hence, we call it the "*fixed basis regression*" model.

If we restrict the basis to $\phi_m(x_i, w_0) = 1$ and $\phi_m(x_i, w_1) = x_i$, $(y(x_i) = w_0 + w_1 x_i)$ then it is a *Linear regression model*. If $\phi_m$ is a monomial basis, then it is nothing but *Polynomial regression*. If $w$ is random and $c$ is learnable, it represents an *Extreme Learning Machine model*. If the $\phi_m$ were Fourier basis, it would

represent *Random Fourier features model.* If $w$ was learnable by the model, it would be a single-layer feed-forward neural network (without bias term). We showcase the relation in increasing level of complexity in Figure 1. Hence, the understanding of the model considered in this article leads us to gain insights into many other related models.

Note, as $w$ are fixed, the number of learnable parameters in this model are $p$. Moreover, the choice of $\Phi_m$ can be different from the true $\Phi_t$ and, normally, $\bar{c}$ is different from $\bar{c}^{true}$ and it is to be learned. Crucially, the choice of model parameters ($p$) can also be different to the number of features in the true function ($d$). This is because both $\Phi_t$ and $d$ (and also $\bar{c}^{true}$) are unknown to the learner, in general. In some examples though $d$ can be known. For instance, for the problem of signal reconstruction (a type of approximation problem where both the true basis $\Phi_t$ and the band-limit $d$ is known) in signal processing the band-limit $d$ is e.g. 22 kHz for audio signals.

### 2.3 Identifiability of a model

It is important to distinguish between identifiability of a model and non-identifiability of model dependent on mapping from parameter space to the model function class. Let us define global identifiability of a model mathematically.

Let $\mathcal{X}$ be the input domain and let $\mathcal{F} \subseteq \{f : \mathcal{X} \to \mathbb{R}\}$ be a hypothesis (model) class indexed by a parameter space $C \subseteq \mathbb{R}^p$. Let

$$\mathcal{M} : C \to \mathcal{F}, \qquad \bar{c} \mapsto f_{\bar{c}}$$

denote the parameter-to-function map induced by the model. If the mapping is many-to-one it is called as non-identifiable model.

**Definition 2.1** (Identifiability). The model is said to be *identifiable* if the map $\mathcal{M}$ is injective, i.e.

$$f_{\bar{c}_1}(x) = f_{\bar{c}_2}(x) \quad \forall x \in \mathcal{X} \implies \bar{c}_1 = \bar{c}_2. \tag{5}$$

Equivalently, distinct parameters correspond to distinct functions in the hypothesis class. If the injectivity condition is violated, it is a functionally-non-identifiable Chatterjee and Sudijono (2025); Zhao et al. (2025).

The non-identifiability of a basis expansion parametric model is dependent on the symmetries of the model basis used. We add example of identifiability and prove that fixed-basis regression model is identifiable in the Appendix A.

Let us define data dependent identifiablity here

Given a finite training set $\{x_i\}_{i=1}^n \subset \mathcal{X}$, define the restriction map

$$\mathcal{M}_n : \Theta \to \mathbb{R}^n, \qquad \theta \mapsto \big(f_\theta(x_1), \ldots, f_\theta(x_n)\big).$$

The model is said to be *data identifiable* if the map $\mathcal{M}_n$ is injective, otherwise it is *data non-identifiable*

This notion of identifiability is about mapping from parameter space to discrete points (training inputs and predicted outputs) on model function class. Hence, it becomes data dependent identifiability in comparison to equation 5.

A model can be identifiable but it can be data non-identifiable. For example, polynomial regression is model-identifiable in under-parameterized regime, however in over-parameterized regime there are infinitely many possible parameters vectors which pass exactly through the training data points making the model, data-unidentifiable.

### 2.4 Training

Once, the model basis is chosen we should perform rescaling of the domain of the dataset to the domain in which the model basis is defined. This process is a homeomorphism and does not affect the structure and

| | Pseudo-Inverse $(\Phi^{\#}_{p\times n})$ | Identity |
|---|---|---|
| $(n > p)$ | $(\Phi^{\dagger}\Phi)^{-1}_{p\times p}\Phi^{\dagger}_{p\times n}$ | $\Phi^{\#}\Phi = I$ |
| $(n = p)$ | $\Phi^{-1}_{n=p\times n=p}$ | $\Phi^{-1}(x)\Phi(x) = \Phi\Phi^{-1} = I$ |
| $(n < p)$ | $\Phi^{\dagger}_{p\times n}(\Phi\Phi^{\dagger})^{-1}_{n\times n}$ | $\Phi\Phi^{\#} = I$ |

Table 1: This table gives expression of Moore-Penrose Pseudo Inverse in different regions, and their properties. † represents transpose-conjugate of the matrix.

topological properties of the true function. For example, Legendre basis is only defined between the domain $x \in [-1.0, 1.0]$.

Now, the task of training is to find the optimal coefficients of the model called $\bar{c}^{opt}$ such that we fit the model $y$ to resemble the true function $g$. This can be done by various methods, like using pseudo-inverse or with optimization methods like gradient descent. Pseudo-Inverse methods provide unique solutions, i.e the solution which is closest to the origin. However, they are only theoretically possible to be applied in cases where the model is linear in parameters. Moreover, in case of large datasets or many parameters, pseudo-inverse methods are not practical due to computational limitations. On the other hand, gradient descent methods solve the issues affecting the pseudo-inverse methods. However, they are iterative and slow and depend highly on initialization sometimes making them as a random guess, without using known structure of the solution. Moreover, they can get stuck in local minima of train loss if the function is non-convex and avoid reaching global minima, which is both boon and bane as we discuss in section 6. As our model is linear in parameters and we are trying to understand the theoretical basis of generalization, we first focus on the pseudo-inverse solution, which always leads us to zero train loss parameter solutions which have minimum 2 norm.

Using the Pseudo-inverse method the optimal set of coefficients to fit the model to true function is

$$\bar{c}^{opt} = \Phi^{train^{\#}}_{m}\bar{y}^{train} = \Phi^{train^{\#}}_{m}(\Phi^{train}_{t}\bar{c}^{true} + \bar{E}) \tag{6}$$

We used Equation 2, Equation 3 and Equation 4. The symbol $\#$ denotes the pseudo-inverse, and we remind that $\bar{c}^{opt} \in \mathbb{R}^{p\times 1}$ and $\Phi^{train^{\#}}_{m} \in \mathbb{R}^{p\times n_{train}}$. There are three different cases of pseudo-inverse that may arise depending on its dimension, shown in Table 1, considering that $\Phi$ is always full rank, which is guaranteed if $w$ are unique. The expression in the second equality shows the mathematical relation between the true coefficients $\bar{c}^{true}$ and the trained coefficients $\bar{c}^{opt}$. In general, these can only be equal if the model basis functions and the true basis functions agree and there is no noise.

We can check if the learned $\bar{c}^{opt}$ effectively predicts the training points via

$$\hat{\bar{y}}^{train} = \Phi^{train}_{m}\bar{c}^{opt} = \Phi^{train}_{m}\Phi^{train^{\#}}_{m}(\Phi^{train}_{t}\bar{c}^{true} + \bar{E}), \tag{7}$$

where we used Equation 6 and Equation 2 and we denoted with $\hat{\bar{y}}^{train} \in \mathbb{R}^{n_{train}}$ the predicted vector. From now on, we will use the hat symbol ( ˆ ) to denote predicted quantities.

It is well known that good training can sometimes be misleading for approximation task. What matters in a good machine learning model for approximation is that it performs well on unknown points. These unknown points can be of two types, one inside domain and other outside domain as discussed earlier. As we will prove and demonstrate, the latter can only be achieved by strict generalization, while the previous one can be achieved by both strict and weak generalization. We will define these in a mathematical form once we discuss more nuances.

Along these lines, consider unknown data points that were not used in training the model

$$\hat{\bar{y}}^{test} = \Phi^{test}_{m}\bar{c}^{opt} = \Phi^{test}_{m}\Phi^{train^{\#}}_{m}(\Phi^{train}_{t}\bar{c}^{true} + \bar{E}). \tag{8}$$

where $\hat{\bar{y}}^{test} \in \mathbb{R}^{n_{test}}$ is the predicted vector. Here, $\Phi^{test} \in \mathbb{R}^{n_{test}\times p}$. Note that $\Phi^{test}_{m}$ is defined on test inputs lying in the range $[a, b]$, which can be different from $[a_{train}, b_{train}]$. We consider $n_{test} \gg n_{train}$ so as to

theoretically better evaluate the capabilities of the model. If the test points are uniformly distributed and their number tends to infinity, the test loss converges, in a functional sense, to the true loss. This limiting quantity is entirely model-dependent and characterizes the intrinsic limitations of the model: it measures the error that remains even in the presence of perfect, noiseless data, reflecting what the model is fundamentally incapable of representing.

Now, we can check the training and testing performance of the model in unknown parts by using the metric called "residual" as given below

$$\bar{R}^{train/test} = \hat{\bar{y}}^{train/test} - \bar{g}^{train/test} \tag{9}$$

Note that in case of training the second term is Equation 4 while in case of testing it is nothing but the true function at the test points without any sampling error, as discussed earlier for theoretical purposes.

Having said this, let us classify generalization. This nuanced understanding of generalization, leads us to understand the conundrum around generalization capabilities of over-parameterized machine learning models. It also showcases the limits of the over-parameterized models in terms of approximation.

**Definition 2.2** (Strict Generalization)**.** We say that **strict generalization** is achieved if and only if

1. **Zero Test Residual (Functional Exactness).**

$$\bar{R}_{\text{test}} = \bar{0}.$$

2. **Recovery of True Coefficients up to Symmetry (Structural Exactness).**

$$\bar{c}_{\text{opt}} \in \mathcal{O}(\bar{c}_{\text{true,pad}}).$$

Here, $G$ denotes the group of functional symmetries of the model acting on the coefficient space, such that

$$\Phi_m \bar{c} = \Phi_m(g \cdot \bar{c}), \qquad \forall g \in G \qquad \forall x \in [a,b]$$

The corresponding functional equivalence class (orbit) of a coefficient vector $\bar{c}$ is defined as

$$\mathcal{O}(\bar{c}) = \{ \, g \cdot \bar{c} \mid g \in G \, \}.$$

For identifiable models, this symmetry group is trivial, implying $g = I$.

In the above definition for fixed basis regression models

$$\bar{c}_{p \times 1}^{true,pad} = \begin{pmatrix} \bar{c}_{d \times 1}^{true} \\ 0_{(p-d) \times 1} \end{pmatrix}_{p \times 1} \tag{10}$$

Since the objective is to learn the true coefficients of the underlying function, such generalization enables extrapolation, that is, predicting function behavior outside the training domain.

However, we must be cautious: extrapolation is only reliable in cases where the function exhibits repeating structure across the domain. For example, linear and periodic functions possess globally consistent patterns, which makes them inherently suitable for extrapolation. In contrast, non-linear, non-periodic functions typically lack such regularity, so extrapolation is only an approximation and becomes increasingly unreliable the farther we move from the training domain. In mathematical terms to represent the same function in a different domain we need a different set of $\bar{c}^{true}$. It changes the farther we go away from the training domain.

However, generalization can also be achieved without learning the true coefficients (features), as we will showcase later in this article.

|  | Parametrization | Sampling | Expressivity |
|---|---|---|---|
| Under | $p < n$ | $n < d$ | $p < d$ |
| Threshold | $p = n$ | $n = d$ | $p = d$ |
| Over | $p > n$ | $n > d$ | $p > d$ |

Table 2: Table depicting various regions of the learning space.

**Definition 2.3.** We define **weak generalization** when $\bar{R}^{test} = 0$ even if we do not learn the true coefficients or its' non-identifiable counterparts.

We showcase that such generalization then is only restricted to the training domain (i.e. in-domain regression) and we cannot extrapolate in such a case if we do not learn the true coefficients.

Let us now understand the criteria required for strict generalization and weak generalization. Also, let us prove/disprove which type of generalization we can observe in over-parameterized regime with experimental proof.

## 3 Strict Generalization

To understand the generalization properties in more detail we need to define new thresholds apart from the interpolation threshold (also called "parametrization threshold" in Table 2). We call the condition where $n = d$, the "sampling threshold" (ST) and the condition where $p = d$, the "expressivity threshold" (ET) and, as already mentioned, when $n = p$ we call it the "interpolation threshold" (IT). See also Table 2 for a summary of the various regimes. As we sweep the number of parameters $p$ while keeping $n$ constant we pass through the ET, but the condition that we are below ST or not is decided when we choose the number of training data points ($n$), and it is not visible on the learning curve plots.

Let us derive the conditions needed for strict generalization and understand the regions in which we can obtain it.

**Theorem 3.1.** *The necessary conditions for $\bar{c}^{opt} = \bar{c}^{true,pad}$ in the fixed basis regression model trained using pseudo-inverse method are*

1. *Zero sampling-noise contribution:* $\Phi_m^{train\#} \bar{E} = \bar{0}$

2. *Enough Expressivity:* $p \geq d$.

3. *Sampling sufficiency:* $n \geq d$.

4. *Under-parameterization:* $n \geq p$.

5. *Span-inclusion:* $\text{Span}\{\phi_t(x_i; w)\}_{i=1}^n \subseteq \text{Span}\{\phi_m(x_i; w)\}_{i=1}^n$.

The most important implication in terms of approximation capabilities in over-parameterized regimes is that we cannot have strict generalization in an over-parametrized regime with the fixed basis regression model.

*Proof.* Let us start with Equation 10. We know that from Equation 6 and Equation 2 that

$$\bar{c}^{opt} = \Phi_m^{train\#} \Phi_t^{train} \bar{c}^{true} + \Phi_m^{train\#} \bar{E} \tag{11}$$

For Equation 10 to be true we need either $\Phi_m^{\#} \bar{E} = \bar{0}$ or the sampling noise $\bar{E} = \bar{0}$. This gives us the first criterion. So Equation 10 reduces to obtaining conditions for

$$\bar{c}^{opt} = \Phi_m^{train\#} \Phi_t^{train} \bar{c}^{true} = \bar{c}^{true,pad}. \tag{12}$$

This implies that the necessary condition for it to be true is

$$\Phi_m^{train\#} \Phi_t^{train} = \left( \frac{I_{d \times d}}{0_{(p-d) \times d}} \right)_{p \times d} = I_{p \times d}^{pad}. \tag{13}$$

As $\text{Rank}(I^{pad}) = d$, we see that we should have $p \geq d$. This gives us the second criteria, that the model has to be expressive enough, and it is decided by the "expressive threshold".

Let us write

$$\Phi_m^{train\#} = \left( \frac{A_{d \times n}}{B_{(p-d) \times n}} \right)_{p \times n}, \tag{14}$$

so according to Equation 13 we should have

$$A \Phi_t^{train} = I_{d \times d}, \tag{15}$$

$$B \Phi_t^{train} = 0_{(p-d) \times d}. \tag{16}$$

The first equation implies that $A$ must be left inverse of $\Phi_t^{train}$, only then can it be satisfied. As left inverse only exists if $n \geq d$, i.e. the number of training points needs to cross the sampling threshold, for $\bar{c}^{opt} = \bar{c}^{true,pad}$. This leads us to the third condition for obtaining strict generalization. This criteria is similar to the "Shannon-Nyquist sampling theorem" in signal reconstruction theory and Sampling complexity in PAC theory Kearns and Vazirani (1994); Ehrenfeucht et al. (1989). Moreover,

$$\text{Im}(\Phi_t^{train}) \subseteq \text{Ker}(B), \tag{17}$$

namely the column space of $\Phi_t^{train}$ is contained in the null-space/kernel of B. A null-space of B is a set of all vectors $\bar{v}$ such that $B\bar{v} = \bar{0}$. Note, that $\bar{v}$ need not be zero vector always, for it to be true. Now, let $\text{Rank}(B) = r$, then by the Rank-Nullity Theorem,

$$\text{Rank}(B) + \text{Nullity}(B) = r + \dim(\text{Ker}(B)) = \dim(\text{Dom}(B)) = n \tag{18}$$

Here Dom means "Domain" and dim is to represent "dimensions". This gives us $\dim(\ker(B)) = n - r$, i.e. any of set of vectors in $\text{Ker}(B)$ can have atmost $n - r$ linearly independent vectors. Now according to Equation 17, $\dim(\text{Im}(\Phi_t(x)) \leq \dim(\text{Ker}(B))$. This means $d \leq n - r$. Now, the rank $r$ can be either $n$ or $p - d$, as we are considering full rank assumption. Suppose $r = n$, then we get $d \leq 0$, which is not possible as $d$ is an integer greater than or equal to 1, hence, for strict generalization $r$ cannot be equal to $n$. Suppose we use $r = p - d$, then $d \leq n - (p - d)$, this gives us the criteria that $n \geq p$, for Equation 16 to be satisfied. Satisfying all these conditions leads us to obtain $\bar{c}^{opt} = \bar{c}^{true,pad}$.

Merely satisfying Equation 10 does not lead to strict generalization, we also need according to Equation 9, $\bar{R}^{test} = \Phi_m^{test} \bar{c}^{opt} - \Phi_t^{test} \bar{c}^{true} = \Phi_m^{test} \bar{c}^{true,pad} - \Phi_t^{test} \bar{c}^{true} = \bar{0}$, for this along with Equation 10 we need $\Phi_m^{train} = (\Phi_t^{train} | \Phi_{n \times (p-d)}')_{n \times p}$ (i.e. $\text{Span}\{\phi_t^{train}(x; w)\} \subseteq \text{Span}\{\phi_m^{test}(x; w)\}$).

If we achieve all the conditions mentioned above and learn the true coefficients, then $\bar{R}^{test} \approx \bar{0}$, leading to strict generalization. However, we may not be able to learn it if the size of training domain is very small compared to the test domain for a general set of functions (except linear and periodic functions), as we discussed in previous section. This finishes the proof. $\square$

Let us now justify the previous result using examples. Before going ahead we need to emphasize that to compare behavior across datasets and scales we will use "normalized root mean square" rather than "root mean square" error. Note that it does not change the behavior of the error except that it rescales the error for comparison. The equation for normalized root mean square is given as

$$\text{NRMSE}_{\text{train}} = \frac{\sqrt{\frac{1}{n_{train}} \sum_{i=1}^{n} R_i^{train^2}}}{(y_{max} - y_{min})^{train}}, \tag{19}$$

where $\bar{R}^{train}$ is the residual in the equation Equation 9. Similarly we can define the $\text{NRMSE}_{test}$ for the test error.

We illustrate our result using one-dimensional continuous function constructed from a Chebyshev polynomial basis of maximum degree $d = 13$. The training dataset consists of $n = 26$ noise-free points, restricted in the domain $[-0.5, 0.5]$. We intend to perform out-of-domain approximation in the region $[-1.0, 1.0]$. Note that

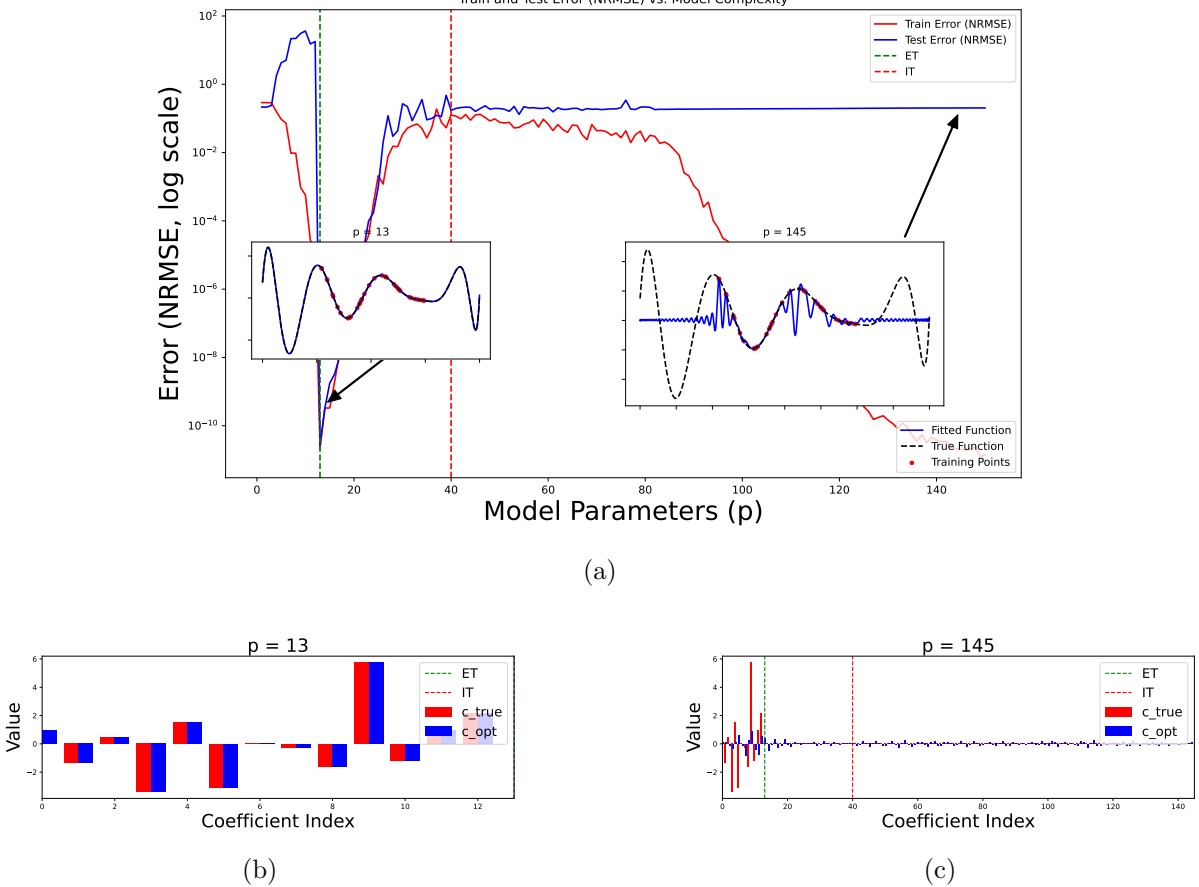

(a)

(b)                                                                      (c)

Figure 2: (a) NRMSE (log-scale) v/s model parameters ($p$). We are interested in out-of-domain approximation and demonstrating the observations of Theorem 3.1. The true function is generated from linear combination of Chebyshev polynomial basis. It can however be any other type of 1-d continuous function. The order of Chebyshev polynomial is $d = 13$. We consider $n = 40$ randomly spaced noise-less training points, restricted to the domain $[-0.5, 0.5]$ and test the approximation in the domain $[-1.0, 1.0]$. We are above the sampling threshold ($(n = 40) \geq (d = 13)$). Inset shows approximation at $p = 13$ and $p = 145$. We observe that $p = d = 13$ the model extrapolates, while at $p = 145$ (over-parameterized regime) the extrapolation capability is lost. (b–c) Histogram comparing $\bar{c}^{opt}$ and $\bar{c}^{true}$ at $p = 13$ and $p = 145$. At $p = 13$, the model learns the true coefficients as expected from Theorem 3.1 and looses it in the over-parameterized regime at $p = 145$.

we are well above the sampling threshold ($n = d$). The model is built using a Chebyshev basis, identical to that of the true function. In particular, as we increase the complexity of the model ($p$), and when it is equal to the highest degree of the true function ($d$) ($p = d$), all the conditions required for strict generalization are met, as stated in Theorem 3.1. In this regime, the model achieves perfect extrapolation. The corresponding results are presented in Figure 2. We plot the normalized root mean square error (NRMSE) as a function of the model complexity $p$. Additionally, we compare the true coefficient vector ($\bar{c}^{\text{true}}$) with the learned coefficients ($\bar{c}^{\text{opt}}$) both at $p = d$ and at an over-parameterized setting $p \gg d$. The associated fitted functions at these complexities are also shown to illustrate the behavior of the model. We can also observe that we do not achieve strict generalization and hence out-of-domain approximation in the over-parametrized regime, which is one of the main results of the theorem above.

We are still unaware of the precise structure of the matrix $\Phi'_{n \times (p-d)}$ that is required for achieving *sustained strict generalization* in the regime $d < p < n$. While we observe generalization exactly at $p = d$ satisfying the conditions above, increasing model complexity beyond this point does not guarantee that the learned coefficients will match the true coefficients—unless additional conditions are satisfied.

We refer to this intermediate region, where $d < p < n$ but strict generalization can still occur under specific structural constraints, as the "extrapolation regime". This regime represents a phase in the learning process where *strict generalization* is possible but not guaranteed, depending on the nature of $\Phi'$ and the alignment between the model and the true function.

**Corollary 3.2.** *Strict generalization in the extrapolation regime ($d < p < n$) occurs only for an orthogonal basis in the fixed basis regression model, even if all the conditions proved in Theorem 3.1 are satisfied.*

Check Appendix C for the proof.

We should note that when we are dealing with discrete orthogonal polynomial basis; they are usually orthogonal with respect to some weighting function ($W$) which depends on the input ($x$), i.e.

$$\sum_{i=0}^{n-1} \phi^a(x_i)\phi^b(x_i)W(x_i) = c_b\delta_{a,b} \tag{20}$$

Where $c_b$ is a real number. Some orthogonal polynomials (like Legendre) have a weighting function equal to 1, even the basis like Fourier basis have a weighting function equal to 1. However, orthogonal polynomials like Chebyshev have a non-constant weighting function. Monomial basis do not form an orthogonal polynomial basis. Hence we cannot expect strict generalization for them in the described region, unless we consider it into account. We can always absorb the weighting function inside the orthogonal polynomial basis and avoid the weighting function, when choosing a model.

Let us demonstrate this understanding in Figure 3 while approximating the function outside the training domain in the region $[-1.0, 1.0]$. We consider a function generated from the Fourier basis. The highest frequency of the basis considered is $f_{max} = 6$, this leads to the true complexity of the function to $d = 2f_{max} + 1 = 13$. We consider a model with a Fourier basis for regression and consider $n = 35$ noiseless training points which are randomly spaced in the domain $[-0.5, 0.5]$. This way we are above the sampling threshold (i.e. $n = d$). As we increase the model-complexity ($p$) and we surpass the expressive threshold ($p = d$), we satisfy all the conditions needed for strict generalization. However, as we increase the model complexity in the region ($d < p \leq n$), because the Fourier basis is orthogonal to each other, we can sustain the strict generalization in the extrapolation regime discussed in Corollary Corollary 3.2.

To conclude the discussion on strict generalization, let us observe, what happens when we violate one of the assumptions in Theorem 3.1. Let us consider that we do not sample enough Figure 4(a) i.e. do not surpass the sampling threshold ($n = d$). We consider another case where we do not choose a right basis Figure 4(b). What happens if we violate both Figure 4(c). All of the scenarios show that we achieve Double Descent but we do not generalize well in the over-parameterized regime if we use Pseudo-inverse method for training the model. This also showcases that the usual bias-variance tradeoff behavior applies to strict generalization and models trained using pseudo-inverse method. What happens with gradient descent is discussed in section 6.

## 4 Weak Generalization

We previously demonstrated that strict generalization is not achievable in the over-parameterized regime, which in turn led us to conclude that even out-of-domain approximation fails under such conditions. Now, let us explore whether in-domain approximation is feasible in the over-parameterized setting.

Can this be achieved without recovering the true coefficients (or features) of the target function? Is it necessary to adhere to the specific conditions we derived earlier for in-domain approximation, or can we succeed without them? Furthermore, in the previous section, we observed that strict generalization requires the model's basis to closely match that of the target function. Does this constraint also apply here? Or is it sufficient to use a more flexible basis—such as one with universal approximation capabilities?

Ultimately, this discussion helps us understand the core puzzle: Why is it that an over-parameterized model, despite being under-determined can still approximate a target function effectively within the training domain?

Let us recall "Weierstrass approximation theorem" from approximation theory Rudin (1976). It states that

**Theorem 4.1.** *For any real continuous function $g(x)$ defined on a closed and bounded interval $[a, b]$, there exist polynomials $y_p(x)$, where $p$ is highest degree in the polynomial, such that $\lim_{p \to \infty} y_p(x) = g(x)$ uniformly for all points in the domain.*

In the language of Machine learning, $y$ represents the model we use for learning and $g$ is the true function. As the order of the polynomial basis increase to infinity, (in our fixed basis regression model, it is the number of parameters) $\lim_{p \to \infty} y_p(x) = g(x)$ for such functions; $\bar{R}^{test} = \bar{0}$, provided that sampling noise is negligible ($\bar{E} = \bar{0}$). Note that this theorem is applicable when the domain of the polynomial is same as the domain of the function to be approximated. This theorem strengthens the notion of generalization in over-parametrized regime for in-domain generalization of continuous functions. Moreover, we recall from approximation theory that Bernstein polynomial basis achieves this irrespective of the target function (it has to be continuous). However, other polynomial basis like Chebyshev basis or Legendre basis cannot achieve this.

Note, this theorem is different from the Universal approximation theorem Cybenko (1989), which does not concern itself with the number of parameters/neurons used to approximate a target function.

It shows us that it is not surprising that we observe generalization in over-parameterized regime, despite it contradicting the conventional notion of "statistical learning theory".

### 4.1 Heuristic proof of weak generalization in over-parameterized regime using Bernstein basis

Let us study the notable case of using Bernstein basis for fixed basis regression model. The proof of Weierstrass approximation is available in textbooks, but we prove it again here in the machine learning perspective and showcase that it (in-domain) approximates continuous functions in over-parameterized regimes, irrespective of learning the true coefficients of the true function and irrespective to the type of the target function (except that it has to be continuous and lie in the range $[0, 1]$).

The Bernstein basis is defined as

$$\Phi_{m_{i,j}} = b_{j,p-1}(x_i) = \binom{p-1}{j} x_i^j (1 - x_i)^{p-1-j}, \tag{21}$$

where here $x \in [0, 1]$. Note, through out this section, we will consider the training domain as same as test domain. It can be observed that the Bernstein basis is nothing but the probability mass function of a binomial distribution. This plays an important role in its being a universal model basis for in-domain generalization. The feature matrix of this basis is

$$\Phi_m = \begin{bmatrix} b_{0,p-1}(x_0) & b_{1,p-1}(x_0) & \cdots & b_{p-1,p-1}(x_0) \\ \vdots & & & \\ b_{0,p-1}(x_{n-1}) & b_{1,p-1}(x_{n-1}) & \cdots & b_{p-1,p-1}(x_{n-1}) \end{bmatrix}_{n \times p}. \tag{22}$$

By De-Moivre Laplace theorem, for $p \to \infty$ such polynomials can be approximated by a Gaussian distribution

$$\lim_{p \to \infty} b_{j,p-1}(x_i) = \frac{e^{-(j-\mu_i)^2/2\sigma_i^2}}{\sqrt{2\pi\sigma_i^2}}, \tag{23}$$

where $\mu_i = (p-1)x_i$ and $\sigma_i^2 = (p-1)x_i(1-x_i)$. We demonstrate this behavior in Figure 5, for various sampling schemes. We can write

$$\lim_{p \to \infty} \Phi_{m_{i,j}} \approx a_i \delta_{j, \lfloor (p-1)x_i \rfloor}, \tag{24}$$

where $a_i$ is some positive real number and $\lfloor \ \rfloor$, represent the floor function, as the index has to be an integer. So,

$$\lim_{p \to \infty} \Phi_m \bar{c} \approx \lim_{p \to \infty} \sum_{j=0}^{p-1} a_i \, \delta_{\lfloor (p-1)x_i \rfloor, j} \, c_j, \quad \text{where } \delta \in \mathbb{R}^{n \times p}, \ c \in \mathbb{R}^p. \tag{25}$$

The right inverse of $\delta$ matrix given here, when $p > n$, is approximately equal to $\delta^T$. We showcase this in the Figure 6.

The optimal coefficients using the training inputs can then be written as using the Pseudo-inverse methods as

$$\lim_{p \to \infty} c^{opt} \approx a_i^{-1} \delta_{j, \lfloor (p-1)x_i^{train} \rfloor} y(x_i^{train}), \tag{26}$$

$$\lim_{p \to \infty} c_j^{opt} \approx a_j^{-1} y(\frac{j}{p-1}). \tag{27}$$

This indicates that $c^{opt}$ passes through the sampled points. We showcase this behavior in Figure 7.

Now definition of test residual Equation 9 and Equation 24

$$\bar{R}^{test} = \hat{\bar{y}}^{test} - \bar{g}^{test} = \Phi_m^{test} \bar{c}^{opt} - \bar{g}^{test} \tag{28}$$

$$= a_k (\delta_{\lfloor (p-1)x_k^{test} \rfloor, j}) y(\frac{j}{(p-1)}) a_j^{-1} - \bar{g}^{test} = y(x_k^{test}) - g(x_k^{test}). \tag{29}$$

If there was no error in sampling the training points, it will not propagate into the test prediction and $\bar{y}^{test}$ becomes equal to $\bar{g}^{test}$, making the residual equal to zero (otherwise non-zero according to the noise in sampling).

We illustrate this phenomenon in Figure 8. The true underlying function is constructed using a Legendre basis of maximum degree $d = 17$. Using $n = 50$ training points sampled uniformly from the domain $[0, 1]$, we fit a fixed-basis regression model. We use the domain because, Bernstein basis is restricted to this domain. When evaluating this model in the same domain using an over-parameterized representation via the Bernstein basis, we observe that the model continues to generalize well despite the increased model complexity. Backed by the proof above we can see that this behavior is a direct consequence of the Weierstrass approximation theorem, stated above. While this result is fundamental in approximation theory, it is often overlooked in discussions within the machine learning community. Recognizing its implications sheds light on why in-domain approximation of function in over-parameterized regimes should not be surprising.

This behavior is a direct consequence of the Weierstrass approximation theorem, stated above. While this result is fundamental in approximation theory, it is often overlooked in discussions within the machine learning community. Recognizing its implications sheds light on why in-domain approximation of function in over-parameterized regimes should not be surprising. Figure 11 uses the same function as in Figure 10. We can observe that Chebyshev polynomial basis does not gener- alize well in the over-parametrized regime. Showcasing that Weierstrass approximation showcases that generalization in over-parameterized regimes for in-domain approximation of continuous functions can be achieved, but not with any type of polynomial.

Figure 9 uses the same function as in Figure 8. We can observe that Chebyshev polynomial basis does not generalize well in the over-parametrized regime. Showcasing that Weierstrass approximation showcases that generalization in over-parameterized regimes for in-domain approximation of continuous functions can be achieved, but not with any type of polynomial.

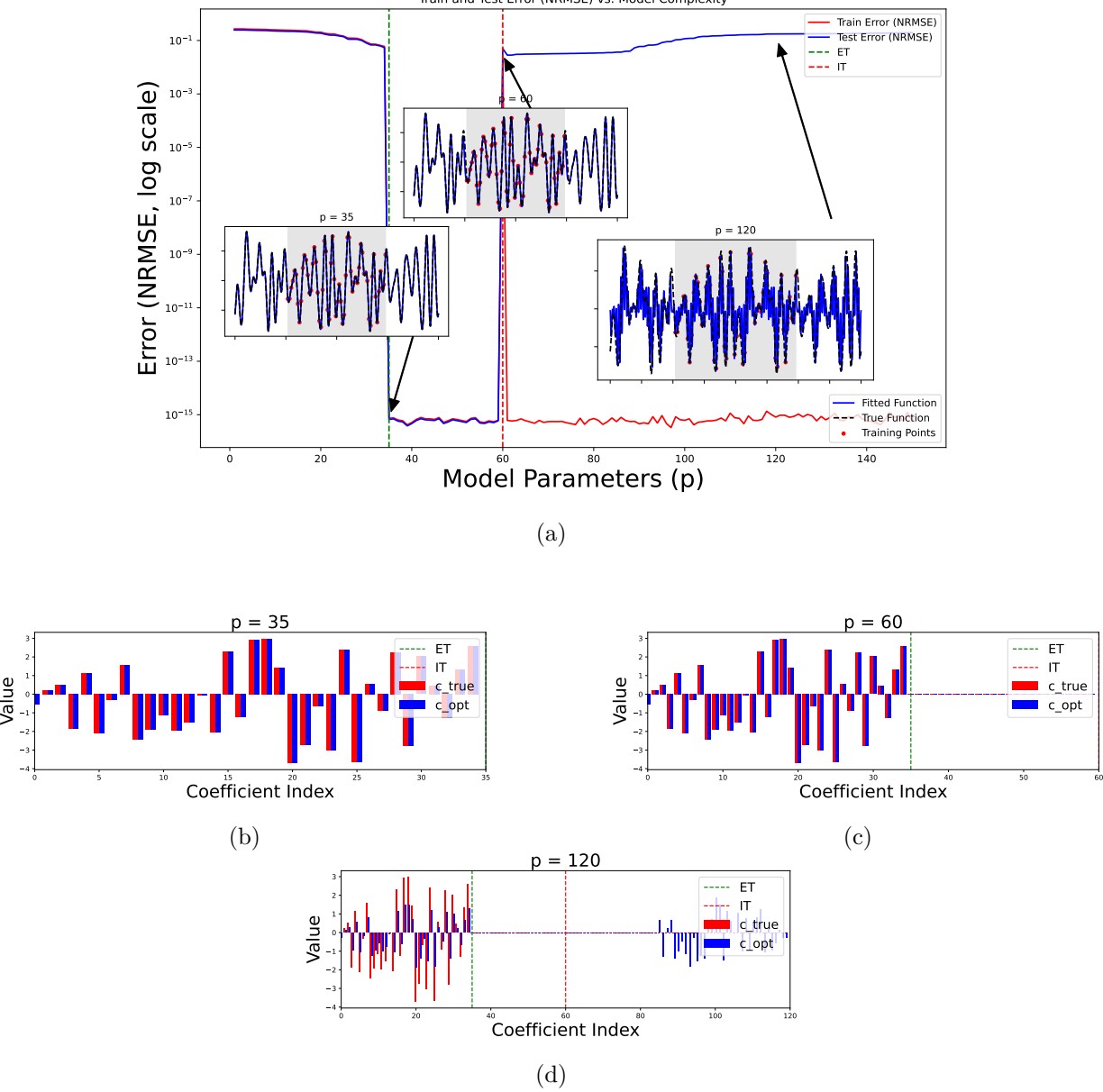

(a)

(b)

(c)

(d)

Figure 3: We showcase sustained strict generalization in the "extrapolation regime" for out-of-domain approximation. Main plot considers NRMSE (log-scale) v/s model parameters ($p$). The true function is a periodic function which can be written using Fourier basis which is orthogonal basis. In this case $d = 2 * f_{max} + 1 = 17 * 2 + 1 = 35$. We consider $n = 60$ uniformly spaced training points, restricted to the domain $[-0.48, 0.48]$ and the test domain is $[-1.0, 1.0]$. We are above the sampling threshold $((n = 60) \geq (d = 35))$. (a) Showcases error ($NRMSE$) v/s model complexity as we increase the number of parameter $p$. Inset plots show the fit at $p = 35, 60, 120$. We can observe that we loose the extrapolation capability at $p = 120$ as expected from the Theorem 3.1. The sustained strict generalization between $ET$ and $IT$ is guaranteed in this case for Fourier basis (as it is orthogonal) by Corollary 3.2. (b–d) Histograms comparing $\bar{c}^{opt}$ and $\bar{c}^{true}$. They demonstrate the coefficients are exactly learned at $p = 35$ and $p = 60$ (which is in the region $d < p \leq n$ ), but it is lost in over-parameterized regime $p = 120$ as expected by Theorem 3.1.

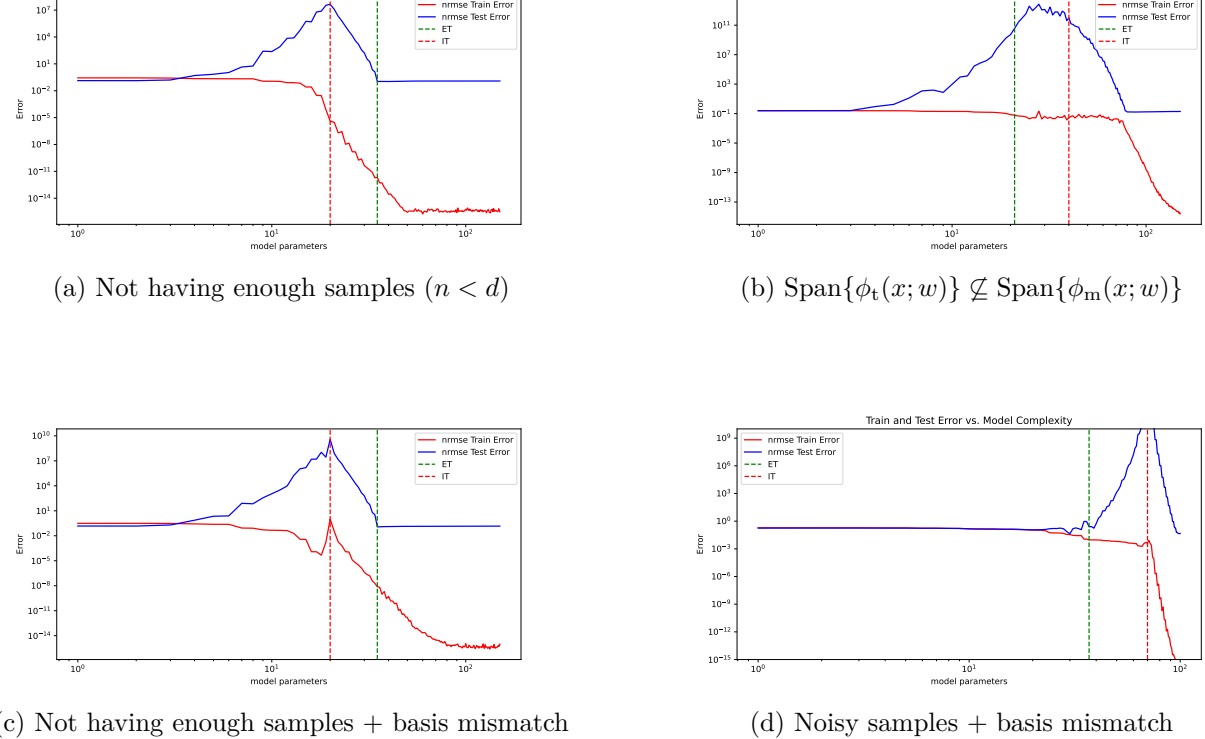

(a) Not having enough samples $(n < d)$

(b) $\mathrm{Span}\{\phi_{\mathrm{t}}(x; w)\} \not\subseteq \mathrm{Span}\{\phi_{\mathrm{m}}(x; w)\}$

(c) Not having enough samples + basis mismatch

(d) Noisy samples + basis mismatch

Figure 4: Illustration of different types of model violations. In (d) we observe triple descent behavior at $(n = d)$ as observed in d'Ascoli et al. (2020).

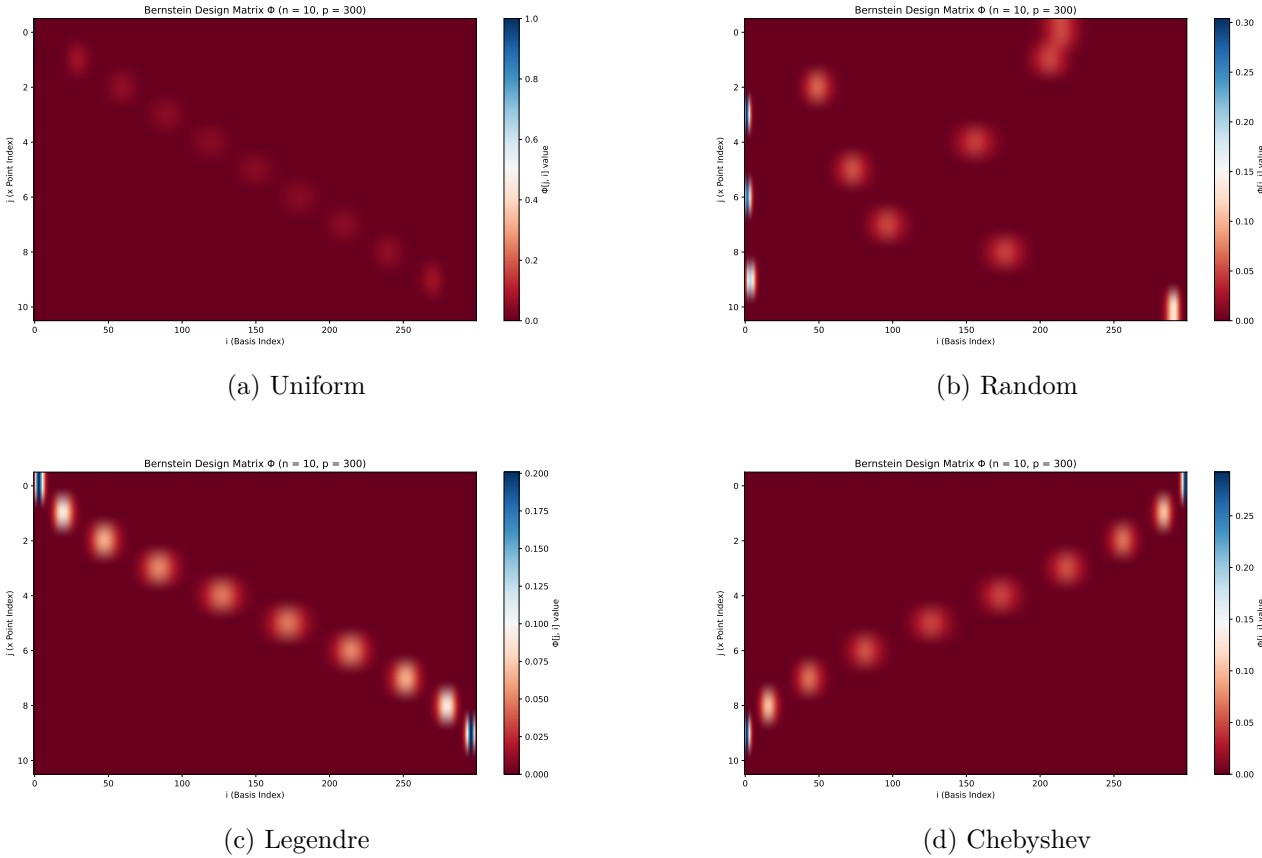

(a) Uniform

(b) Random

(c) Legendre

(d) Chebyshev

Figure 5: We plot the values of the Bernstein basis design matrix. It can be seen that we observe the gaussian distribution with mean at points given by $\lim_{p \to \infty} b_{j,p-1}(x_i) = \frac{e^{-(j-\mu_i)^2/2\sigma_i^2}}{\sqrt{2\pi\sigma_i^2}}$

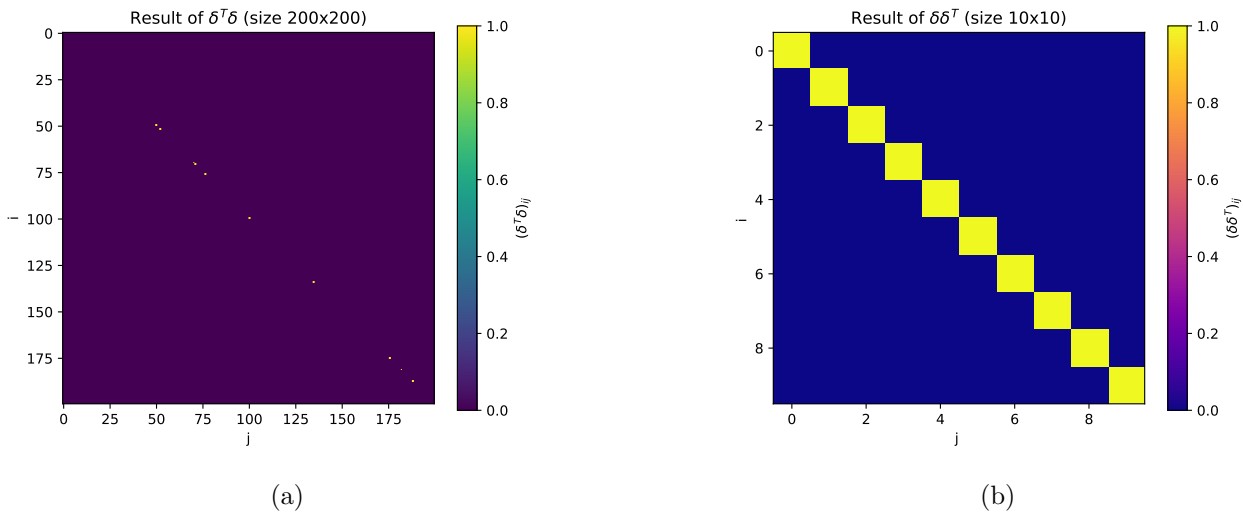

(a)

(b)

Figure 6: We consider a $\delta_{n \times p}$ ($n = 10$ and $p = 200$) matrix defined in Equation 24, where $x_i$ are ($n = 10$) generated randomly in the domain $[0, 1]$. (b) Showcases that the right inverse of such a matrix is its' transpose when $p \geq n$. (a) Showcases the tranpose of $\delta$ matrix is not its left inverse.

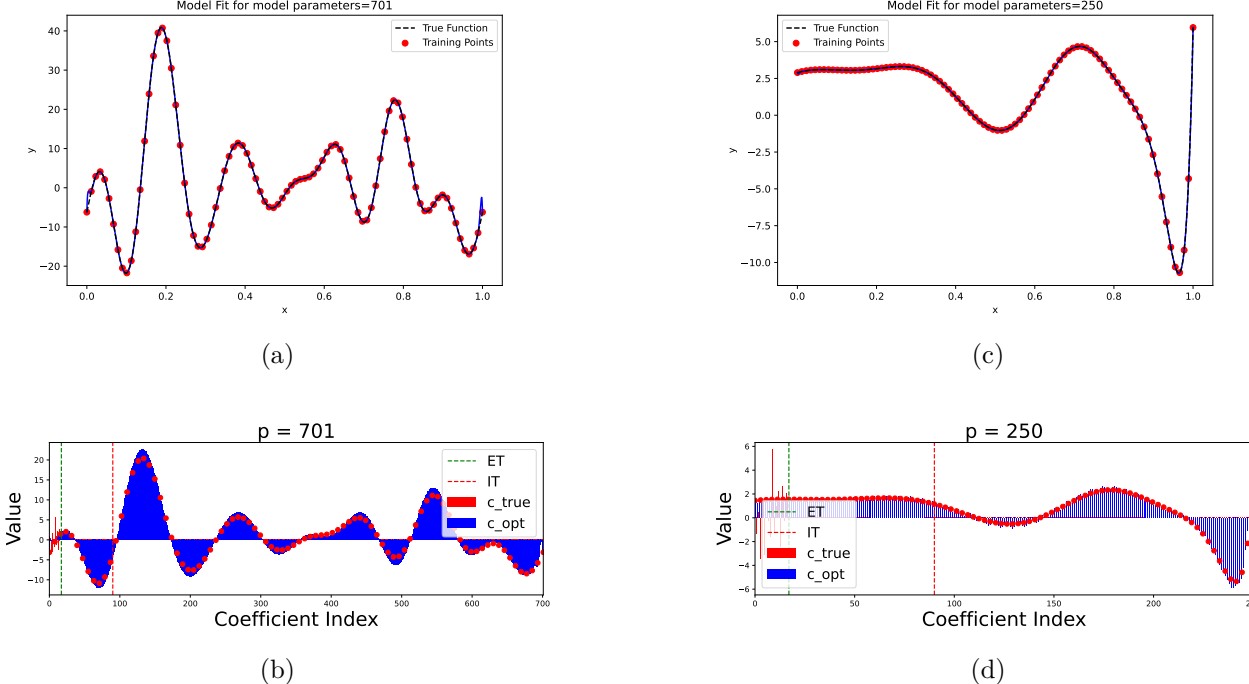

(a)

(c)

(b)

(d)

Figure 7: In these plots we showcase that the coefficients become approximately equivalent to the sampled training data points, when we use Bernstein basis for regression. (a) Shows the in-domain approximation of function generated using Fourier basis, with highest frequency equal to 17 and 90 training data points. The points are sampled randomly. The plot is generated at 700 model parameters. (b) Shows the histogram of true and learned coefficients. It can be seen that their values are equivalent to that of the training data points. (c) Shows the in-domain approximation of function generated using Legendre basis, with highest frequency equal to 17 and 50 training data points. The plot is generated at 150 model parameters. (d) Shows the histogram of true and learned coefficients. It can be seen again that their values are equivalent to that of the training data points. In both (b) and (d) plots, we should point out that the true coefficients are restricted behind the expressive threshold and are not visible as their values are way less than the learned coefficients.

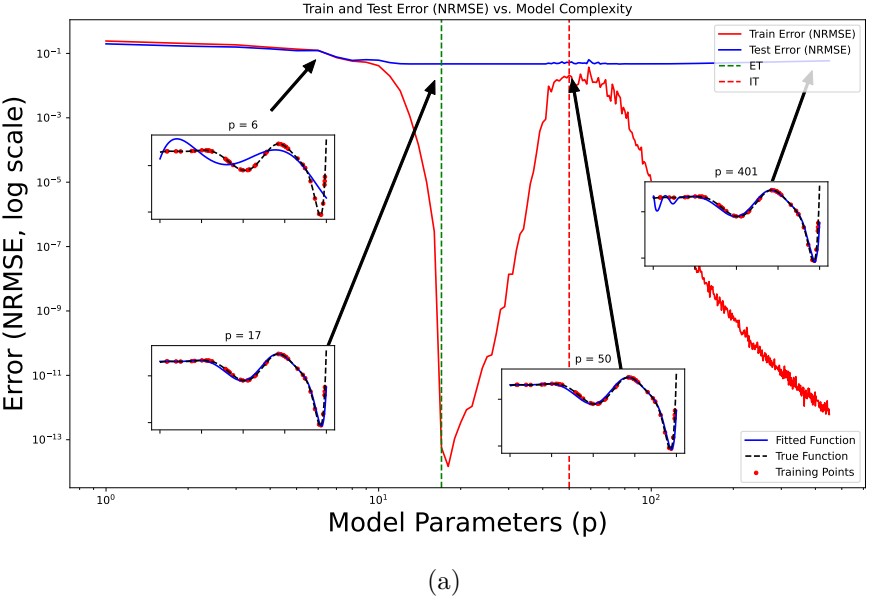

(a)

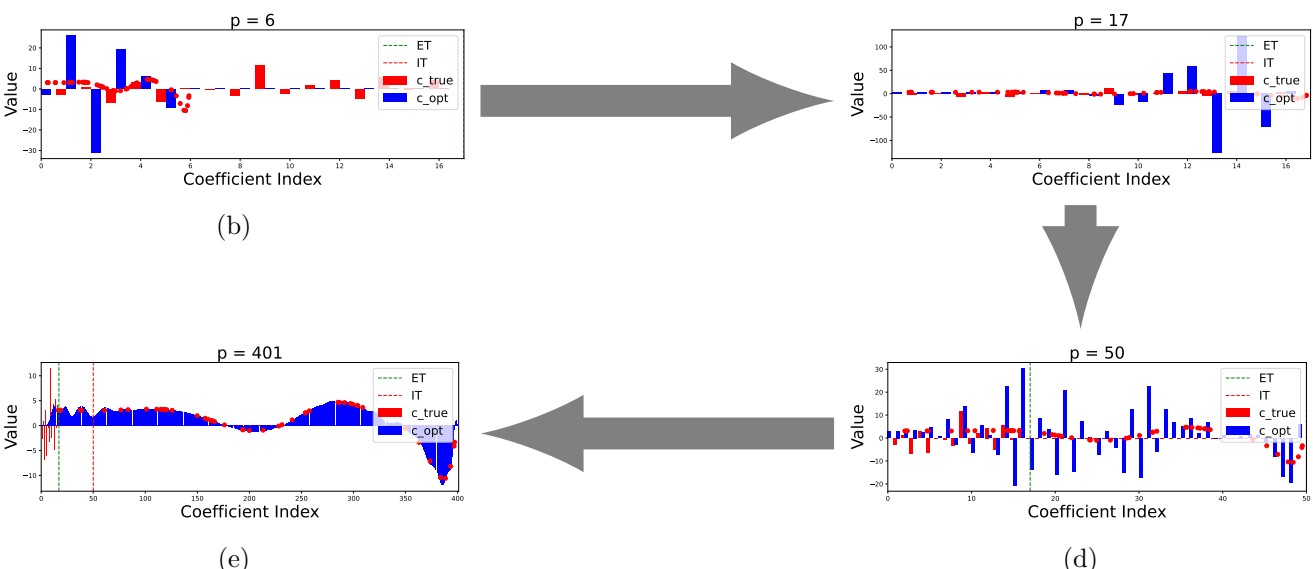

(b)                                                          (e)                                                          (d)

Figure 8: (a) In this figure we plot NRMSE error v/s Model complexity, in log scale. The problem is of approximating a function generated using Legendre basis of highest degree $d = 17$ in the domain $[0, 1]$. We sample $n = 50$ samples at random points. We use Bernstein basis for defining the model. The figures in the inset showcase the approximations over various model complexities. It can be seen that the model generalizes well even beyond over-parameterized regime. This is a demonstration of Weierstrass approximation theorem, which strengthens our observation of phenomenon like benign overfitting and generalization in over-parameterized regime. (b) In this subfigure we showcase histogram of learned coefficients ($c^{opt}$) and true coefficients ($c^{true}$) superimposed by the values of the training data points, at various model complexities. We can observe the property of learned coefficients when using Bernstein coefficients in the histograms at over-parameterized regime.

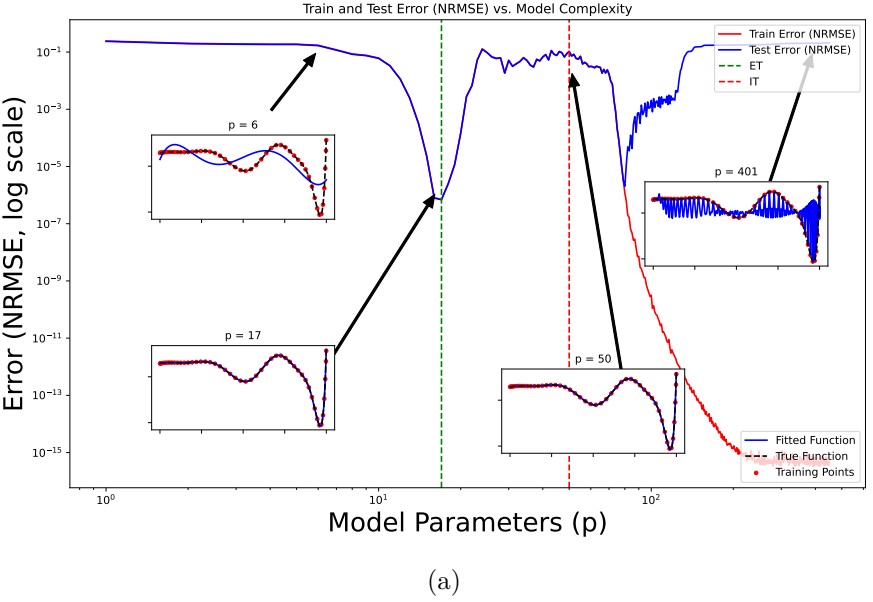

(a)

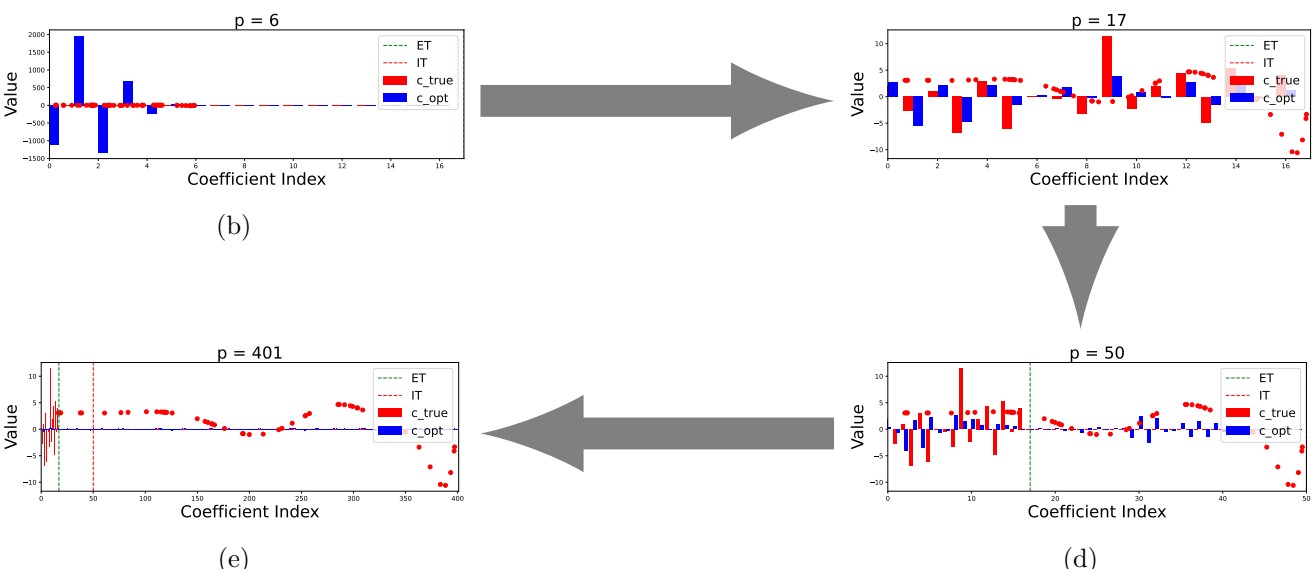

(b)

(e)

(d)

Figure 9: (a) In this figure we plot NRMSE error v/s Model complexity, in log scale. The problem is of approximating a function generated using Legendre basis of highest degree $d = 17$ in the domain $[0, 1]$. We sample $n = 50$ samples at random points. We use Chebyshev basis for defining the model, rather than Legendre. The figures in the inset showcase the approximations over various model complexities. It can be seen that this time the model does not generalize well beyond over-parameterized regime, unlike in the case of Bernstein basis. This showcases that not all polynomials can achieve the approximation capability in the over-parametrized regime. (b) In this subfigure we showcase histogram of learned coefficients ($c^{opt}$) and true coefficients ($c^{true}$) superimposed by the values of the training data points, at various model complexities. We can observe that the learned coefficients ($c_{opt}$) do not match the value of the training points unlike in the case of Bernstein basis.

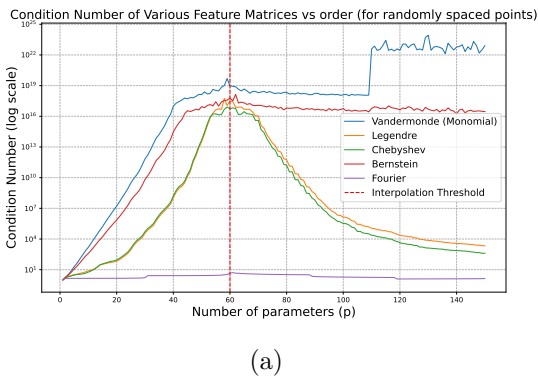
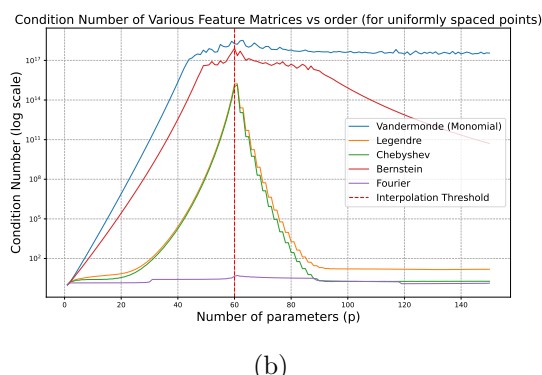

(a)                                                                                    (b)

Figure 10: We consider various feature matrices used in the model and plot their condition number v/s parameters (highest degree of the basis) in the model. We consider $n = 60$ training points which are randomly spaced in (a) and uniformly spaced in (b). The red dashed line represents the interpolation threshold. The plot shows Fourier basis to be most stable while Monomial basis to be ill-conditioned even after the interpolation threshold. We can observe double descent behavior in the condition number for other bases.

## 5 Importance of stability and noise sensitivity

In Theorem 3.1 we sidelined the noise in sampling, let us understand the effect of noise in estimating the true coefficients. This is necessary for getting a complete picture.

According to Equation 11, the second term $\Phi_m^{train^{\#}} \bar{E}$ is the error in the learning of true coefficients if there is error in the inputs themselves. It can be shown that the ratio of the relative error in learned coefficients to the relative error in training data is bounded by the condition number Strang (2012) of the feature matrix itself. For derivation check Appendix D

$$\frac{\|\Phi_m^{train^{\#}} \bar{E}\|}{\|\bar{E}\|} \frac{\|\bar{y}^{train}\|}{\|\Phi_m^{train^{\#}} \bar{y}^{train}\|} \leq \kappa(\Phi_m^{train}). \tag{30}$$

Let us plot the condition number for various model feature matrices, in their respective domains they are defined in, with randomly as well as uniformly spaced inputs $x^{train}$ in that domain Figure 10.

The error part in the test residual $\bar{R}_{test}$ due to sampling error in the training data is $\Phi_m^{test} \Phi_m^{train^{\#}}$. As $\Phi_m^{train^{\#}}$ appears in the error in the test residual, and $\kappa(\Phi_m^{train^{\#}}) = \kappa(\Phi_m^{train})$ we currently only care about its' condition number, by exploiting the sub-multiplicativity property of the condition numbers.

Let us understand the result in detail. It shows that the maximum of the ratio of the relative error in the learned coefficients to the relative error in the training data is highly regular for the Fourier feature matrix, while for the Monomial feature matrix it is highest, irrespective of the sampling choice. Moreover, we see that for the case of monomials it does not show significant double descent, while for others it is significant. We surprisingly observe double descent in the condition numbers of these feature matrices, just like the observation of test errors. A similar observation was also reported for *Random Matrices* and *Radial Kernel* in Poggio et al. (2019).

However, we should note that despite the model improving its condition number in over-parameterized, we cannot learn the true coefficients according to Theorem 3.1, in that regime. The condition number of the feature matrix is only important until and unless the conditions of Theorem 3.1 are met for out-of-domain approximation tasks.

Moreover, in the case of in-domain approximation, we observe that despite Chebyshev basis or Fourier features being highly conditioned in over-parameterized regimes, the condition number does not predict the

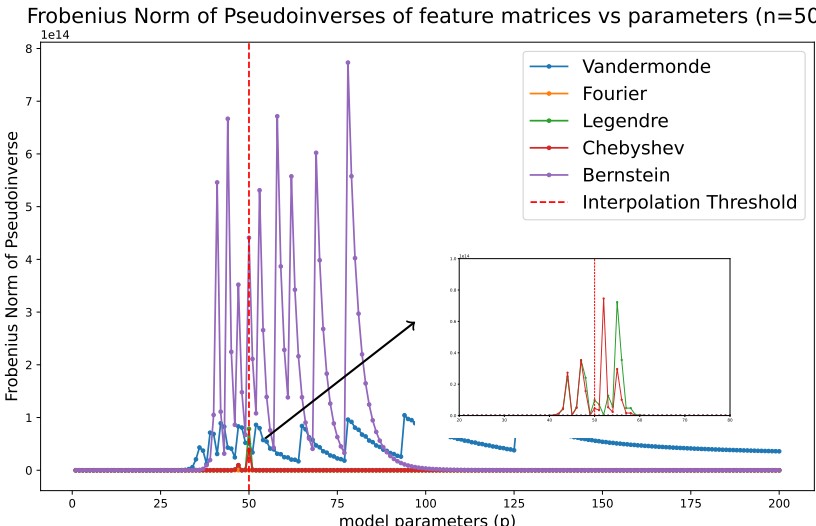

Figure 11: Frobenius norm of the pseudo-inverse of the feature matrices of the model v/s model parameters. The inputs are randomly sampled in the respective domains. The Frobenius norm measures the expected variance in the output with respect to error in the input, if it is a gaussian noise. We sample points It can be seen that it also showcases double descent behavior.

test error behavior. We can observe that a model with Bernstein features is able to approximate well in the over-parameterized regime Figure 8 rather than the Chebyshev features Figure 9. This shows that the choice of basis for approximation is more important than the condition number of the feature matrices, both for in-domain and out-of-domain approximation tasks. Condition numbers, alternatively, only decide the effect of noise in training data on the approximation. Moreover, the ill-conditioning of the model can be regularized by different techniques once the model basis is chosen for specific task, taking into account the condition number of the feature matrix for that number of parameters and training data.

Condition number of the Hessian matrix of the loss function can affect the convergence properties while training using gradient descent, however we do not delve into this in this article Recht (2013), as we have used closed form solution using pseudo-inverse.

Let us now calculate the variance in the learned coefficients in comparison to the noise in the training data. If $\bar{E} \sim \mathcal{N}(0, \epsilon \mathbb{I})$ (i.e. Gaussian noise) and if $\bar{v} = \Phi_m^{train^{\#}} \bar{E}$ then

$$\mathbb{E}(\|\bar{v}\|^2) = \epsilon^2 Tr(\Phi_m^{train^{\#}}(\Phi_m^{train^{\#}})^T) = \epsilon^2 \|\Phi_m^{train^{\#}}\|_F^2 = \epsilon^2 \sum_{i=1}^{r} \sigma_i^2, \tag{31}$$

where $\|\Phi_m^{train^{\#}}\|_F$ is known as the *Frobenius Norm* of the matrix and it is equal to trace or root of sum of the square of singular values of that matrix Strang (2012). In contrast to the condition number we now have an idea of how much training data noise is mapped to learned coefficients. Let us find it for various features Figure 11.

The stability of the model through condition number and its relation to double descent is studied in Poggio et al. (2019); Chen and Schaeffer (2021); Rangamani et al. (2020).

# 6 Insights into the idea of Implicit Regularization of Gradient Descent

Initializing close to zero leads us to minimum L2 norm solution as it is closest to zero vector. Minimum L1 norm cannot be close to zero as there can be one element which is very high making the length of the vector very high but it can be sparser than the min L2 norm solution.

## 6.1 Classification of parameters

In this section we formally classify parameters into various sets, from our understanding from previous sections. We will define these sets for model-identifiable models only.

Let us define the set of all parameter vectors in the parameter space of the model as $C \subseteq \mathbb{R}^p$. Note that the dimensions of this parameter space is different than the dimensions of true parameter space for a identifiable model, let us denote it by $C_t \subseteq \mathbb{R}^d$, hence $\bar{c}^{true} \in C_t$.

Let us denote true loss as $L_{true}$, i.e. the loss considering we have infinite noiseless-test points on the domain on which the ground truth is defined. Let us not give it any functional form for generality.

- True loss minimizer set
  $C_{true-min} \equiv \arg\min_{\bar{c} \in C} L_{true}(\bar{c})$ This set represents the best approximation of $g(x)$, the ground truth by the chosen model on all points on which the ground truth is defined.

- Zero True loss parameters
  $C_{sg} \equiv \{\bar{c} \in C : L_{true}(\bar{c}) = 0\}$. This is the set of parameters which makes true loss zero, hence in such a case we approximate the function at all points in domain of the true function. This is the case of strict generalization and parameter recovery up to the functional identifiability of the model. For functionally identifiable models, this is a singleton set as there is an injective mapping between parameter space and model function class. In our case in Theorem 3.1 where we train an identifiable model with pseudo-inverse method, it is equal to $\bar{c}^{true,pad}$. This makes its' true loss like a convex bowl, if the model is expressive enough. However, for functionally non-identifiable models (like neural networks), this set is non-singleton and there can be many parameter vectors in this set which can make true loss zero Chatterjee and Sudijono (2025), leading to multiple global minima. For example, one could permute the weights between neurons in each layer and the output variable would still be the same, or multiple periodic symmetry introduced if learnable sin features are introduced with sin activation in neural networks Zhao et al. (2025). If $C_{true-zero} \neq \varnothing$ then $C_{true-zero} = C_{true-min}$ as the minimum of a true loss is zero.

- Training loss minimizer set
  $C_{train-min} \equiv \arg\min_{\bar{c} \in C} L_{train}(\bar{c})$ This is the set of parameters which minimize the training loss. Minimization of training loss is not equivalent to approximation of the ground truth.

- Zero Train loss parameters
  $C_{train-zero} \equiv \{\bar{c} \in C : L_{train}(\bar{c}) = 0\}$. This set is non-singleton for the fixed basis regression model we consider, in the over-parameterized regime. This makes the train loss function have multiple global minima. There are infinitely many possibilities of parameter vectors which can make train loss zero. In fact, $C_{train-zero}$ is an affine subspace of dimension $p-n$ (assuming the design matrix $\Phi$ has full row rank). All these parameter vectors perfectly fit the training data, i.e., for each training input $x_i$, the model prediction equals the training output $y_i$. For existence of this set, conditions of Theorem 3.1 need not be satisfied.

- Zero train loss parameters with minimum 2 norm
  $C_{m2n} \equiv \arg\min_{\bar{c} \in C_{train-zero}} ||\bar{c}||_2$ This is the set of all minimum 2-norm parameter vectors which make train loss zero. This is exactly the solution which a pseudo-inverse method finds for fixed basis regression models. It is also the solutions which vanilla gradient descent finds when initialized at zero vector, with small enough learning rate and sufficient epochs, this property is termed as implicit regularization of gradient descent. This set is singleton for fixed basis regression models.

- Weak generalizing parameters
  Consider a neighborhood around each training point $x_i$ $N_i = \{x : ||x - x_i|| \leq \epsilon\}$, then
  $C_{wg} = \{\bar{c} \in C : y(x) \approx g(x), \forall x \in \bigcup_{i=1}^n N_i\}$ This is the set of all parameters which approximate the ground truth not only at the training point but also in a neighborhood around it. One of the example of existence of such set of parameters is the example of Bernstein basis. Such parameters can exist even if $C_{sg}$ set does not (i.e. conditions of Theorem 3.1) are not met) as demonstrated.

Another such example of parametric methods is that of spline methods and neural networks with ReLU activation Balestriero et al. (2018), where the true parameters are not learned but the function is fitted near the training points. These are the parameter vectors we achieve when we approximate a continuous function using Bernstein basis in section 4. The existence of these parameters is not dependent on the conditions of Theorem 3.1. A rigorous study of existence and properties of such parameters is an open problem.

Let us understand what if the condition of under-parameterization of Theorem 3.1 is violated. $C_{sg}$ is singleton and lies on a hyperplane in the model parameter space of an identifiable model as $\bar{c}_{opt} = \bar{c}^{true,pad}$ provided other conditions of Theorem 3.1 are satisfied.

When we use vanilla gradient descent and initialize model parameters near zero vector, it chooses the minimum 2-norm training solution $C_{m2n}$ as it is closest to the origin, this set is singleton for an identifiable model and is a subset of $C_{train-zero}$. However, $C_{sg}$ which is the true solution is not the same as $C_{m2n}$. This vector exists no matter the size of the model parameter space if other conditions of $Theorem$ 3.1 are satisfied. It can be reached by an optimization solution which chooses the local minima (like vanilla gradient descent) if we initialize our parameters near $C_{sg}$.

Let us demonstrate this Figure 12. The setup in this demonstration is identical to that of Figure 3, except that we now compare pseudo-inverse training with gradient descent initialized close to the true parameter values in the model's parameter space. The configuration in Figure 3 already partially satisfies the conditions of Theorem 3.1. As expected, pseudo-inverse training fails to achieve strict generalization in the over-parameterized regime, whereas gradient descent succeeds when started near the true coefficients $\bar{c}^{true,pad}$.

However, it may remain a random guess in an over-parameterized situation to reach $C_{sg}$, as it needs knowledge of elements of $C_{sg}$ itself. This also showcases that we should avoid using pseudo-inverse methods in over-parameterized regime if we intend to achieve strict generalization.

## 7 Insights into deep learning

Let us understand if neural networks, when they approximate well, do they learn the coefficients upto the functional identifiability (i.e. they obtain strict generalization or not). We use a single hidden layer neural network with one single input and output and '$tanh()$' activation function. It gives us a model function

$$y(x_i^{train}) = \sum_w \phi_m(x_i; w_j, b_j)c_j \tag{32}$$

$$\bar{y}^{train} = \Phi_m^{train}\bar{c} \tag{33}$$

This model is different from Equation 3, in the sense that in this case $w$ is learnable and we have an added parameter $b$ (i.e. bias). In the neural network model sense the weights between the input and the hidden layer are represented by $w$ and the weights between hidden layer and output are represented by $c$. The model now strictly becomes functionally non-identifiable Zhao et al. (2025).

To demonstrate that feedforward neural networks also perform "weak generalization", let us consider a true function defined by Equation 1. Let us consider that it is defined by a Legendre polynomial basis in the domain $[-1, 1]$. Let it be of order $d = 20$. We sample $n = 40$ training points at uniform spacing with Gaussian sampling noise mean 0 and noise standard deviation 0.05. We consider $p = 120$ neurons, leading us to $3 \times p$ unknowns. The true coefficients are randomly generated. Let us perform training from the sampled data using "Gradient Descent" method. The results are shown in Figure 13.

It can be observed that the model generalizes very well in the training domain. However, it does not learn the true coefficients, leading us to conclude that it generalizes weakly according to definition Definition 2.3. Moreover, as we have proven that there is no restriction for weak generalization in over-parameterized regime, we can observe good in-domain approximation. However, according to Theorem 3.1 as we cannot truly learn features in over-parameterized regimes, it hints us towards the explanation that neural networks,

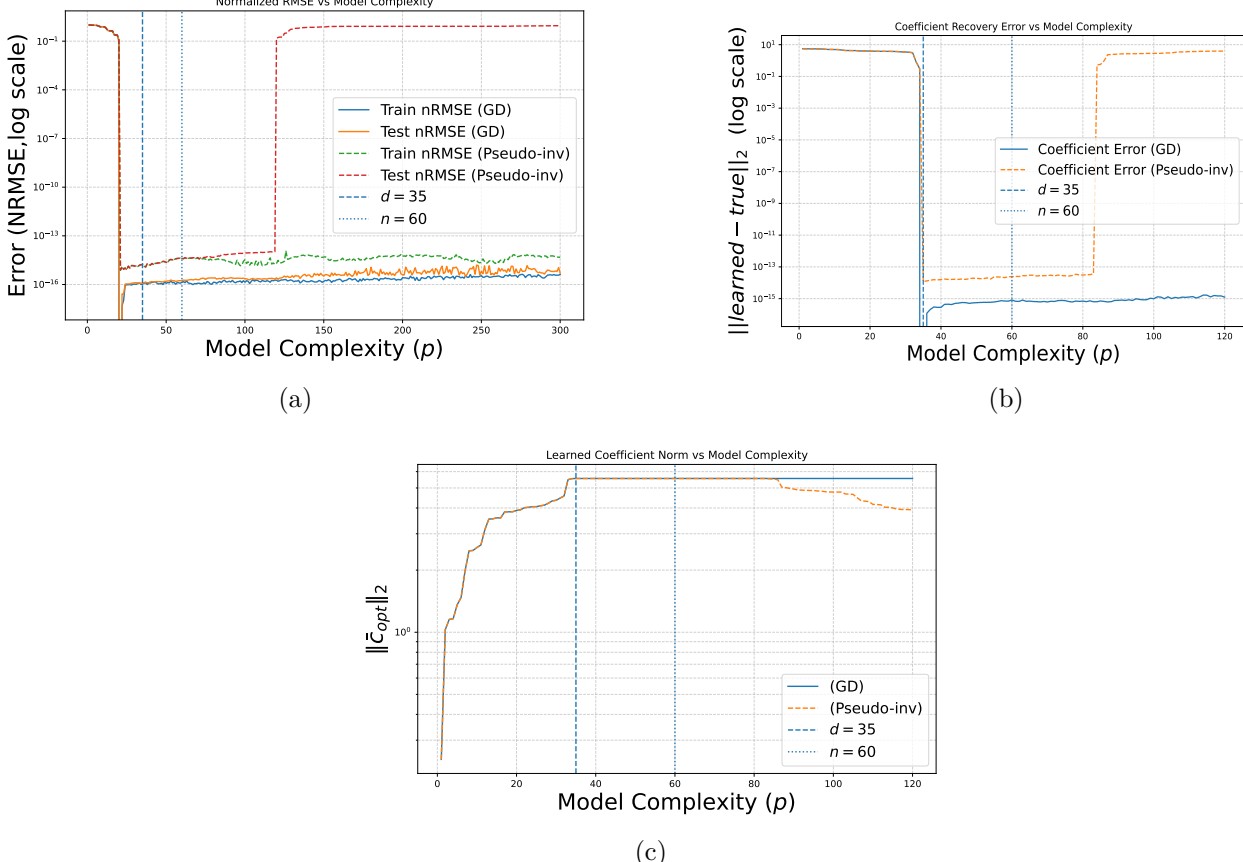

(a)

(b)

(c)

Figure 12: The setup in this demonstration is same as in Figure 3, except that we compare the pseudo-inverse training with gradient descent near true parameter values in the model parameter space. The setup in Figure 3, already partially satisfies the requirements of Theorem 3.1. It can be observed that pseudo-inverse training does not achieve strict generalization in over-parameterized regime however as expected, however gradient descent does, if initialized near true coefficients $\bar{c}^{true,pad}$ in the model parameter space.

in most cases, would not learn the true features of the target function, outside the training domain in the over-parameterized regime, despite generalizing well during in-domain approximation, because according to condition 5 of Theorem 3.1 the activation function of the neural network in most cases does not match the features representing the true function. Neural network models like *SIREN* which use periodic activation functions, strengthen our viewpoint Ziyin et al. (2020); Sitzmann et al. (2020).

Our work is complementary to the study of deep learning models in *infinite-width limit* Jacot et al. (2018). However, our approach is a bottoms-up approach rather than top-down approach of the infinite-width framework. In that framework the neural network turns into a model which performs linear regression with non-linear fixed features. Let us denote the network model with parameters $\theta$ and input $x$ as $f(x; \theta)$. The Taylor expansion around initial parameters $\theta_0$ gives

$$f(x;\theta) = f(x;\theta_0) + \nabla_\theta f(x;\theta_0)^T(\theta - \theta_0) + \frac{1}{2!}(\theta - \theta_0)^T H_f(\tilde{\theta})(\theta - \theta_0) + \cdots \tag{34}$$

in the infinite width limit and with particular initializations the Jacobian term becomes constant in parameters and the Hessian term (and other higher order terms) becomes negligible Lee et al. (2019). This leads us to

$$f(x;\theta) \approx \sum_j \phi_j(x;\theta_0)c_j, \tag{35}$$

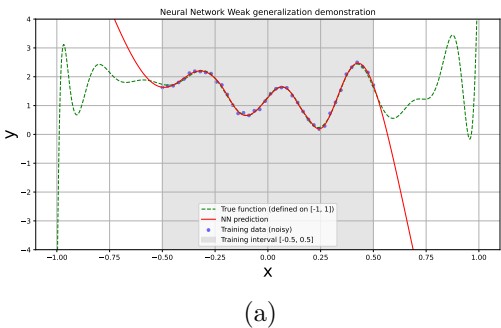

(a)

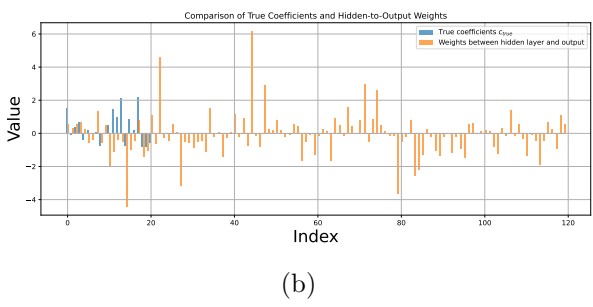

(b)

Figure 13: (a) We consider a true function defined by a Legendre polynomial basis defined in the domain $[-1, 1]$. Let it be of order $d = 20$. We sample $n = 40$ training points in the domain $[-0.5, 0.5]$ at uniform spacing with Gaussian sampling noise mean 0 and noise standard deviation 0.05. The true coefficients are randomly generated. The plot represents the approximation with a single hidden layer neural network with $p = 120$ neurons. Which means the model is over-parameterized. We train using GD and it can be seen that the model fits the function very well (b) It plots the true coefficients with the weights of the neural network between the hidden layer and output. We can observe that they do not match. Indicating that feedforward neural networks generalize weakly.

where the first term in Equation 34 can be considered a constant offset term and the Jacobian can be written as $\nabla_\theta f(x; \theta_0)^T \approx \phi_j(x; \theta_0)$. As $j \to \infty$, this model is always over-parameterized when trying to approximate a function, hence it leads to weak generalization, i.e. the network looses the ability to extrapolate and performs weak generalization.

# 8 Insights into Quantum Machine Learning

Generalization in quantum machine learning is also not completely understood Gil-Fuster et al. (2024). While bounds based on "conventional" statistical learning theory can be applied when the model class is suitably constrained Banchi et al. (2021; 2024); Caro et al. (2021; 2022); Du et al. (2021; 2023), they are known to be vacuous in the over-parametrized regime. On the other hand, benign overfitting and double descent were both observed in the quantum setting Peters and Schuld (2023); Kempkes et al. (2025).

It is important to remark that it is not fully clear yet whether over-parametrization is readily applicable to the quantum setting. This is due to several challenges: on one hand, training gets complicated because of the lack of a simple backpropagation algorithm Abbas et al. (2023); on the other hand, hardware constraints make extremely challenging to significantly increase either the depth or the width of quantum neural networks Schuld and Petruccione (2021). Even to address these challenges, a hardware-friendly approach to achieve over-parametrization with quantum models has been recently proposed Tognini et al. (2025), which is based on a mixture of quantum experts.

We now discuss the applications of our findings for the quantum machine learning community. First, our results are directly applicable to quantum extreme learning machines and quantum reservoir computing Innocenti et al. (2023); Mujal et al. (2021); Nakajima et al. (2019), since both of them can be expressed as in our Equation 3, where $\phi_m(x, w)$ are the "reservoir functions" that, in the quantum setting, are obtained by creating a suitably complex quantum state, e.g. by letting a simple quantum state evolve according to a complex Hamiltonian, and then performing fixed measurements. Here the parameters $w$ define the internal dynamics of the quantum system, e.g. the Hamiltonian parameters, and the measurement settings. Our results can be applied to better understand which reservoir functions can guarantee better generalization and stability against noise.

Another application is for models constructed via classical shadows Jerbi et al. (2024), since some of them can also be expressed as a linear model Equation 3 where the basis functions are obtained by first loading data into quantum states (in a non-linear way) and then measuring certain observables.

Finally we discuss applications for quantum neural networks, which are among the most popular quantum machine learning models Schuld and Petruccione (2021). Even such models can be linearized via natural tangent kernel Liu et al. (2022); Shirai et al. (2024); Girardi and De Palma (2025) and thus be expressed as Equation 3, though the validity of such approximation is not completely understood. General quantum neural networks defined through common quantum gates, even with reuploading layers Pérez-Salinas et al. (2020), can always be expressed as a partial Fourier series Schuld et al. (2021)

$$f(x, \theta) = \sum_{\omega \in \Omega} c_\omega(\theta) e^{i\omega x}, \tag{36}$$

where the expansion coefficients $c_\omega(\theta)$ depend in a complex way on the network parameters $\theta$ (e.g. qubit rotations) and the space of possible frequencies $\Omega$ may grow exponentially with the number of qubits and circuit layers. Because of Equation 36, most common quantum neural networks can be expressed as a linear model in a Fourier-like space, though with expansion coefficients that depend in a non-linear way on the trainable parameters. Nonetheless, quantum neural networks often display generalization properties similar to the ones that we observe with Fourier models, or the related Chebyshev polynomials, e.g. the "spiky" behaviour that may eventually lead to benign overfitting Peters and Schuld (2023).

## 9 Conclusion

Despite observation of Double Descent we could observe that models did not generalize well in the over-parameterized regime. We also try to understand that despite the problem being under-determined in this regime it approximates well. This led us to understand that there are two types of generalization possibilities. One, where we not only limit the residual error to a very small value but also learn the features of the true function globally, we call this as "strict generalization". On the other hand we can achieve very small residual error irrespective of learning the true features of the function to be approximated. We call this as "weak generalization". We argue that while with strict generalization we can achieve out-of-domain approximation of a continuous function, with weak generalization we only achieve in-domain approximation. This distinction is also partially backed by the spline theory of ReLU networks, which showcase that they behave essentially like adaptive linear spline regression models Sahs et al. (2022); Balestriero et al. (2018).

We derive the necessary conditions for strict generalization in fixed basis regression models trained using pseudo-inverse exact methods Theorem 3.1. These methods fall into the category of functionally identifiable models Zhao et al. (2025), and we restrict our analysis to such models only. In doing so, we learn that strict generalization cannot be achieved in the over-parametrized regime for a fixed basis regression model Figure 1, trained using pseudo-inverse methods which usually find the minimum 2 norm training solution. Satisfying, the classical bias-variance trade off observation. However, we showcase why gradient descent can surpass this in probabilistic sense (i.e. dependent on initalization) in section 6. Moreover, we provide a taxonomy of various parameter sets observable in model parameter space. It holds insights into the adaptive basis regression model, which is nothing but a vanilla neural network. We additionally define new thresholds, i.e. "Sampling threshold" (which indicates the minimum number of samples needed in comparison to the highest complexity of the function to be approximated) and also "Expressive threshold" (which indicates the minimum model complexity needed to approximate the true function) for fixed basis regression models. These thresholds bring fresh perspective and easily interpretable versions to understand theory of over-parameterized machine learning.

We then study weak generalization in such models, which can be achieved without satisfying conditions of strict generalization, when performing in-domain approximation specifically. We showcase this using a model which uses Bernstein basis. We remind that this is backed by the Weierstrass approximation theorem, which is well known in approximation theory, but is not studied from classification of generalization behavior and parameter sets as we do in this article.

Later we study the stability and noise sensitivity of the model for various basis. We realize the condition number of the Pseudo-inverse of the feature matrix (worst case amplification of noise) and its Frobenius norm (Expected variance in solution due to Gaussian noise), both showcase double descent. We also notice that the double descent in these metrics need not follow the same dynamics for test residuals. It leads us to showcase that a well-conditioned model is important but it is secondary to having the conditions for generalization being met, i.e. the choice of model basis with respect to the problem. Once the basis is chosen and there is enough expressivity and samples we can regularize the model to make it well-conditioned.

These observations lead us to gain insights in the success and limitations of deep-learning, despite them being over-parameterized. We showcase empirically that feed-forward neural networks achieve weak generalization in most cases (when the activation functions do not match the features of the target function). As weak generalization has no restrictions, unlike strict generalization, to be obtained in the over-parameterized regime, we can observe it. As weak generalization restricts us to in-domain approximations only, we can observe why over-parameterized neural networks in those cases are not capable of feature learning outside training domain.

Our work primarily acts as an important taxonomical and theoretical study rather than a study deriving limits and metrics. The derivations and observations resolve several conundrums by providing simple insights, and leave several other studies open. For example, our work is limited to 1-d continuous functions, though it can be generalized to higher dimensions. We would also need a rigorous exploration of the cases where the function to be approximated is discontinuous. Moreover, we only focus supervised learning model like fixed basis regression model. We choose this model as (a) it is fundamental to Machine learning (b) it is easy to find the closed form solution for this model without getting involved in the issues with optimization problems and (c) it is a special case of adaptive basis regression models, which are nothing but vanilla feed-forward neural networks. Having said this, there are some non-basis expansion models like symbolic regression Dick and Owen (2024) and self-supervised learning models Lupidi et al. (2023), which are speculated to show no double descent behavior. These speculations are empirical, and a rigorous proof is lacking. A rigorous study of the weakly generalizing parameter sets in model parameter space is lacking. Their existence conditions, uniqueness and conditions to converge to these solutions using optimizing algorithms is lacking.

We expect that our work will bring a fresh perspective in the quest to understand generalization in over-parametrized regimes and close the gap on understanding generalization in deep learning and over-parametrized machine learning, thus showing us direction to making even more efficient and interpretable deep learning algorithms.

## Acknowledgements

CW and LB have been supported by the European Union – NextGenerationEU under the National Recovery and Resilience Plan (PNRR) – Mission 4 Education and research – Component 2 From research to business – Investment 1.1 Notice Prin 2022 – DD N. 104 del 2/2/2022, from title "understanding the LEarning process of QUantum Neural networks (LeQun)", proposal code 2022WHZ5XH – CUP J53D23003890006. LB has also been supported the European Union's Horizon Europe research and innovation program under EPIQUE Project GA No. 101135288.

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

## A   Identifiability of Fixed Basis Regression

**Proposition (Identifiability of Fixed Basis Regression)**

Consider the fixed basis regression (FBR) model defined in Equation (3):

$$y(x) = \sum_{j=0}^{p-1} \phi_m(x; w_j)\, c_j, \tag{37}$$

where the basis functions $\{\phi_m(\cdot; w_j)\}_{j=0}^{p-1}$ are fixed and the coefficient vector $\bar{c} \in \mathbb{R}^p$ constitutes the learnable parameters.

Assume that the basis functions are linearly independent on the domain $X$, i.e.

$$\sum_{j=0}^{p-1} a_j \phi_m(x; w_j) = 0 \quad \forall x \in X \implies a_j = 0 \quad \forall j. \tag{38}$$

Then the fixed basis regression model is globally identifiable in the sense of Equation 2.1, when equality of model outputs is considered over the entire domain $X$ (equivalently, in the limit $n \to \infty$).

**Proof**

Recall from Equation 2.1 that a model is identifiable if the parameter-to-function map

$$\mathcal{M} : \Theta \to \mathcal{F}, \qquad \bar{c} \mapsto f_{\bar{c}}(\cdot)$$

is injective. For the fixed basis regression model,

$$f_{\bar{c}}(x) = \sum_{j=0}^{p-1} \phi_m(x; w_j)\, c_j.$$

Assume that there exist two parameter vectors $\bar{c}^{(1)}, \bar{c}^{(2)} \in \mathbb{R}^p$ such that

$$f_{\bar{c}^{(1)}}(x) = f_{\bar{c}^{(2)}}(x) \quad \forall x \in X. \tag{39}$$

Substituting the model definition into Equation equation 39 yields

$$\sum_{j=0}^{p-1} c_j^{(1)} \phi_m(x; w_j) = \sum_{j=0}^{p-1} c_j^{(2)} \phi_m(x; w_j), \quad \forall x \in X.$$

Rearranging terms,

$$\sum_{j=0}^{p-1} \left( c_j^{(1)} - c_j^{(2)} \right) \phi_m(x; w_j) = 0, \quad \forall x \in X. \tag{40}$$

Define $a_j := c_j^{(1)} - c_j^{(2)}$. By the assumed linear independence of the basis functions (Equation equation 38), Equation equation 40 implies

$$a_j = 0 \quad \forall j.$$

Hence,

$$\bar{c}^{(1)} = \bar{c}^{(2)}.$$

Therefore, the parameter-to-function map $\mathcal{M}$ is injective, and the fixed basis regression model is globally identifiable. $\qquad\square$

### Remark (Model vs. Data Identifiability)

Earlier proposition establishes *model identifiability*, which is a property of the mapping from parameter space to function space. This should be distinguished from *data-dependent identifiability* (subsection 6.1), where the mapping is restricted to a finite set of training inputs. In particular, although fixed basis regression is globally identifiable, it may become data non-identifiable in over-parameterized regimes ($p > n$).

### Counterexample: A Non-Identifiable Model

**Example (Two-Layer Neural Network).** Consider a single-hidden-layer neural network with scalar output and no bias:

$$f_\theta(x) = \sum_{j=1}^p a_j\, \sigma(w_j x), \tag{41}$$

where $\theta = \{(a_j, w_j)\}_{j=1}^p$ are the model parameters and $\sigma(\cdot)$ is a nonlinear activation function.

For any permutation $\pi$ of $\{1, \ldots, p\}$, define a new parameter set

$$\tilde{a}_j = a_{\pi(j)}, \qquad \tilde{w}_j = w_{\pi(j)}.$$

Then

$$\sum_{j=1}^p a_j \sigma(w_j x) = \sum_{j=1}^p \tilde{a}_j \sigma(\tilde{w}_j x) \quad \forall x \in X,$$

while $\theta \neq \tilde{\theta}$.

Hence, the parameter-to-function map is many-to-one, and the model is non-identifiable in the sense of Definition 2.1.

The non-identifiability in earlier example arises from intrinsic symmetries in the parameterization. Fixed basis regression does not admit such symmetries, since the basis functions are fixed and linearly independent.

## B  Various feature matrices and sampling schemes

Let us define the various features used in the article, the domain in which they are defined and if they are orthogonal, the weight metric under which the basis is orthogonal.

Using the information above, various feature matrices can be generated. For example for the Legendre polynomial basis, the Feature matrix is given as

$$\Phi = \begin{bmatrix} P_0(x_1) & P_1(x_1) & P_2(x_1) & \cdots & P_{d-1}(x_1) \\ P_0(x_2) & P_1(x_2) & P_2(x_2) & \cdots & P_{d-1}(x_2) \\ \vdots & \vdots & \vdots & \ddots & \vdots \\ P_0(x_n) & P_1(x_n) & P_2(x_n) & \cdots & P_{d-1}(x_n) \end{bmatrix} = \begin{bmatrix} 1 & x_1 & \frac{1}{2}(3x_1^2 - 1) & \frac{1}{2}(5x_1^3 - 3x_1) & \frac{1}{8}(35x_1^4 - 30x_1^2 + 3) & \cdots \\ 1 & x_2 & \frac{1}{2}(3x_2^2 - 1) & \frac{1}{2}(5x_2^3 - 3x_2) & \frac{1}{8}(35x_2^4 - 30x_2^2 + 3) & \cdots \\ \vdots & \vdots & \vdots & \vdots & \vdots & \ddots \\ 1 & x_n & \frac{1}{2}(3x_n^2 - 1) & \frac{1}{2}(5x_n^3 - 3x_n) & \frac{1}{8}(35x_n^4 - 30x_n^2 + 3) & \cdots \end{bmatrix} \tag{42}$$

Similarly, other feature matrices can be generated for the model. from Table 3.

| Property | Legendre Basis | Chebyshev Basis (First Kind) | Fourier Basis (Real Form) | Monomial Basis |
|---|---|---|---|---|
| Definition Domain | $x \in [-1, 1]$ | $x \in [-1, 1]$ | $x \in [0, 1]$ or $[-\pi, \pi]$ (We feature scale to $[-1, 1]$) | $x \in \mathbb{R}$ (We feature scale to $[-1, 1]$) |
| Weight Function | $w(x) = 1$ | $w(x) = \frac{1}{\sqrt{1-x^2}}$ | $w(x) = 1$ | Not orthogonal |
| Basis functions | $P_j(x) = 1, x, \frac{1}{2}(3x^2 - 1), \dots$ | $T_j(x) = 1, x, 2x^2 - 1, \dots$ | $F_j(x) = 1, \cos(2\pi x), \sin(2\pi x) \dots$ | $M_j(x) = 1, x, x^2, \dots$ |

Table 3: Comparison of Legendre, Chebyshev, and Fourier bases

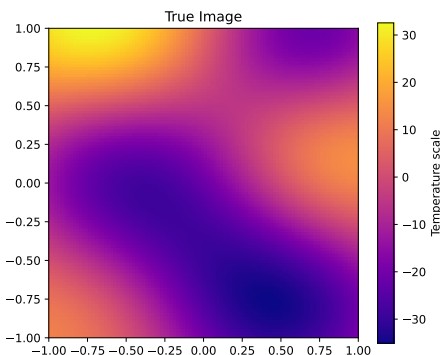

Figure 14: Representation of the temperature field given by Equation 48. We scaled the boundaries to lie between $[-1, 1; -1, 1]$

## C  Proof of Corollary 3.2

We add the proof of Corollary 3.2 here, which is derived from the Theorem 3.1

*Proof.* In general, a vector $\bar{v} \in Ker(M)$, iff $x$ is orthogonal to each row of $M$. This is according to the definition of orthogonality itself.

Now, according to Equation 17 for ensuring sustained strict generalization in the region between $d < p \leq n$ columns of $\Phi_t(x)$ need to be orthogonal to rows of $B$.

That means that '$\Phi_t$' and '$\Phi_m$' need to have an orthogonal basis for strict generalization in the strict generalization regime, for fixed basis regression model.

If the basis are not orthogonal, we cannot extend the strict generalization in this region and it would only occur at $p = d$ i.e. at expressive threshold ( provided that we are above sampling threshold $n \geq d$), provided that other requirements are also fulfilled. □

## D  Proof of how worst case noise amplification is bounded by condition number

The ratio of the relative error in learned coefficients to the relative error in training data is given as

$$\frac{\|\Phi_m^{train^{\#}} \bar{E}\|}{\|\bar{E}\|} \frac{\|\bar{y}^{train}\|}{\|\Phi_m^{train\#} \bar{y}^{train}\|} = \frac{\|\Phi_m^{train^{\#}} \bar{E}\|}{\|\bar{E}\|} \frac{\|\Phi_m^{train} \bar{c}\|}{\|\bar{c}\|} \tag{43}$$

We used the relation Equation 3 in the previous equation. We also know that, $\frac{\|\Phi_m^{train\#}\bar{E}\|}{\|\bar{E}\|} \leq \sigma_{max}(\Phi_m^{train\#})$ and $\frac{\|\Phi_m^{train}\bar{c}\|}{\|\bar{c}\|} \leq \sigma_{max}(\Phi_m^{train})$, where $\sigma_{max}()$ represents the largest singular value of that matrix. The largest simgular value is nothing but the *2-norm* of the matrix. The definition of condition number of $\Phi_m^{train}$ is such that

$$\kappa(\Phi_m^{train}) = \sigma_{max}(\Phi_m^{train})\sigma_{max}(\Phi_m^{train\#}) \tag{44}$$

which leads us to the relation that the ratio of the relative errors in the learned coefficients to the relative error in training data to be bounded by the condition number Equation 30.

## E  Real world example

### E.1  Strict Generalization and surrogate model reconstruction of a temperature field

As an illustrative real-world case of strict generalization, we consider the reconstruction of the steady-state temperature field on a 2-D surface. Temperature field reconstruction refers to estimating the spatial distribution of temperature over a domain using sparse or noisy observations. This arises in simulations, inverse problems, meteorology, and sensor networks, where full field measurements are often impractical.

This is formulated as an inverse problem, where we aim to infer hidden properties of a system from observed measurements, in contrast to a direct problem Nakamura and Potthast (2015). In our setup, we generate a synthetic temperature field (known to us but hidden from the reconstruction model), and task the model with recovering it from partial observations. Moreover, as an added complexity, we pose this as an extrapolation problem where we are unable to get enough observations in some parts of the problem, maybe due to limitations of the instrument.

We use a Chebyshev basis to first approximate the true field. This makes our problem that of a surrogate model Forrester et al. (2008). A surrogate model (or metamodel) is any computationally inexpensive approximation of a more complex, expensive-to-evaluate function. Instead of solving full physics-based PDEs (like Navier-Stokes, heat equation, etc.), we project the field onto a set of Chebyshev polynomials, reducing the problem to a small set of coefficients. We can then reconstruct or predict the field values efficiently using this surrogate.

Let us understand when we can surrogate a model. Let us consider a function $g(x)$ (for example $e^{-x^2}$). This function is not a basis expansion of Chebyshev basis. However, we can approximate it in this basis, calling the expansion as *surrogate* of the actual function. The Chebyshev expansion of a function $g(x)$ on $[-1, 1]$ is

$$g(x) \sim g_{surrogate}(x) = \frac{a_0}{2} + \sum_{i=1}^{\infty} a_i T_i(x), \tag{45}$$

where the coefficients are given by

$$a_i = \frac{2}{\pi} \int_{-1}^{1} \frac{g(x)\,T_i(x)}{\sqrt{1-x^2}}\,dx, \qquad i \geq 0. \tag{46}$$

Using numerical integration we can find the coefficient of expansion as Table 4. Note, the value of the coefficients reduces as the degree $i$ increases of the expansion. In such scenarios we can safely surrogate the ground truth into another basis expansion.

We emphasize that this expansion is not the same as the approximation using regression methods. This is because for evaluating the expansion coefficients we need to know the ground truth as evident in Equation 46. In Pseudo-inverse methods of approximating the ground truth we have no knowledge of the true function.

The example of surrogate models also serves as a demonstration of the limits of extrapolation capabilities of the model, when the model features do not match the exact true features of the ground truth, as given by the conditions in Theorem 3.1.

Table 4: Chebyshev coefficients $a_i$ for $f(x) = e^{-x^2}$ on $[-1, 1]$

| $i$ | $a_i$ |
|---|---|
| 0 | 1.290070540898 |
| 1 | 0.000000000000 |
| 2 | -0.312841606370 |
| 3 | 0.000000000000 |
| 4 | 0.038704115419 |
| 5 | 0.000000000000 |
| 6 | -0.003208683015 |
| 7 | 0.000000000000 |
| 8 | 0.000199919238 |
| 9 | 0.000000000000 |
| 10 | -0.000009975211 |
| 11 | 0.000000000000 |
| 12 | 0.000000415017 |

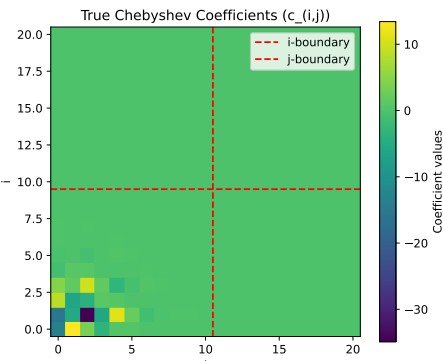

Figure 15: Values of the coefficients in Chebyshev expansion of the true temperature field. We can see that the values of $c_{i,j}$ are approximately equal to zero for values above $i = 10$ and $j = 10$ in the Equation 47. Hence, we consider the true coefficients for surrogate modeling to be $d = 10$ and $e = 10$.

In the 2-d case (assuming that the function represents a temperature field) we can expand the true function as given below

$$T(x, y) \approx \sum_{i=0}^{d} \sum_{j=0}^{e} c_{ij}^{true} \, T_i(x) \, T_j(y) \tag{47}$$

where $T(x, y)$ is approximated temperature field as a function of spatial coordinates $x$ and $y$,. $T_i(x)$, $T_j(y)$ are Chebyshev polynomials of the first kind (orthogonal basis functions). $c_{ij}^{true}$ are Spectral coefficients to be determined from data or projection. $d, e$ are Order (or degree) of the polynomial expansion in $x$ and $y$, respectively.

Now, to evaluate generalization performance in inverse modeling, we construct a synthetic temperature field $T(x, y)$ defined over a 2-D spatial domain. The field is designed to exhibit a combination of smooth global trends, localized sources and sinks, spatial oscillations, and stochastic perturbations, mimicking complex real-world thermodynamic behavior. This field is known to us, but to the approximating algorithm later, we will feed the samples from this field, to showcase our understanding on approximation and strict generalization in the main article.

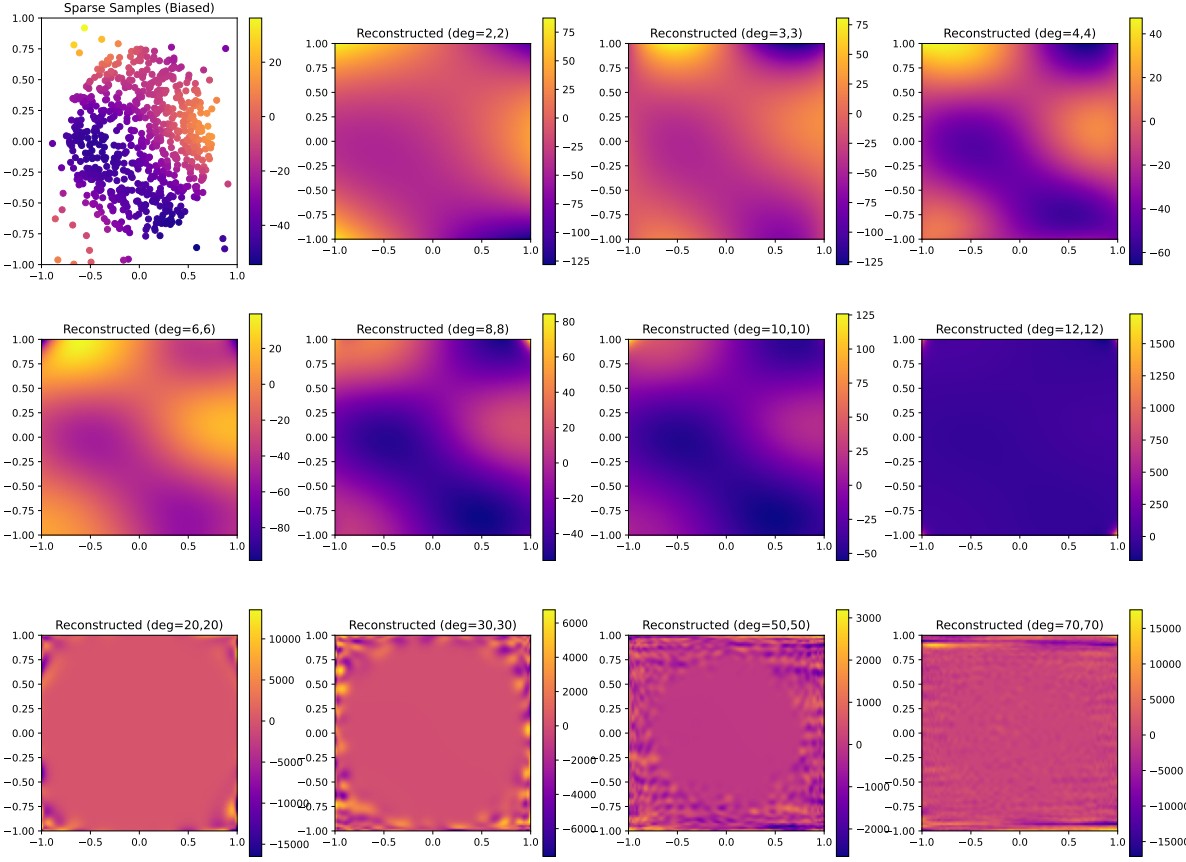

Figure 16: Reconstructed temperature fields from the sampled points. The samples are sparse outside a certain radius in this case it is $r = 0.75$. We can see that at $p, q = 10$, that is the model complexity, we see best reconstruction, the model also extrapolates in the sparsely sampled region. The model looses it reconstruction capability if we increase the parameters further than that.

### E.2   Temperature field generation

The temperature at location $(x, y) \in \mathbb{R}^2$ is given by:

$$
\begin{aligned}
T(x, y) = \ & T_{\text{ambient}} + (T_s - T_{\text{ambient}}) \, e^{-\lambda_c (x^2 + y^2)} \\
& + A_1 \sin(2x) \cos(3y) \, e^{-\lambda_1 ((x-2)^2 + (y-1)^2)} \\
& + A_2 \, e^{-\lambda_2 ((x+3)^2 + (y-4)^2)} \\
& + A_3 \, e^{-\lambda_x x^2 - \lambda_y y^2} \\
& + \alpha_x x + \alpha_y y \\
& + \beta \sin \left( 0.5 x^2 + 0.3 y^2 \right) \\
& + \sigma \cdot \eta(x, y) \\
& + A_{\text{sink}} \, e^{-\lambda_s ((x - x_s)^2 + (y - y_s)^2)}
\end{aligned}
\tag{48}
$$

where:

- $T_{\text{ambient}}$ is the ambient background temperature.

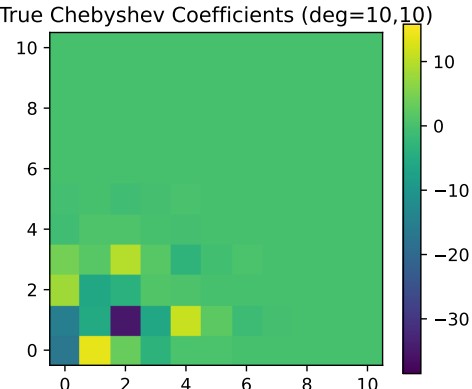 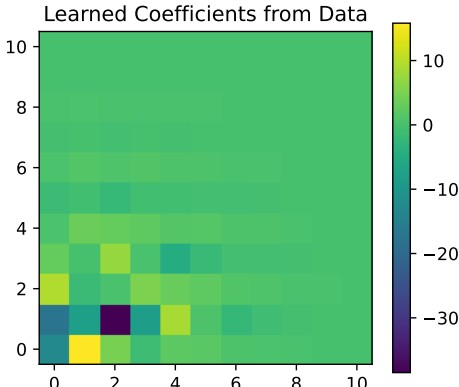

Figure 17: Comparison of true and learned coefficients. We see that the learned coefficients approximately match

- $T_s$ is the strength of the central Gaussian heat source, with decay rate $\lambda_c$.

- The second term models a spatially oscillatory source centered at $(0.3, 1)$, with amplitude $A_1$ and decay $\lambda_1$.

- The third term models a distant source centered at $(-3, 4)$, with amplitude $A_2$.

- The fourth term is an anisotropic Gaussian source with independent decay rates $\lambda_x, \lambda_y$, and amplitude $A_3 \sim \mathcal{U}(0, 1)$.

- The fifth term introduces a linear gradient field with coefficients $\alpha_x, \alpha_y \sim \mathcal{U}(0, 2)$.

- The sixth term is a smooth nonlinear perturbation with amplitude $\beta \sim \mathcal{U}(0, 1)$.

- The seventh term is spatial white noise, where $\eta(x, y) \sim \mathcal{N}(0, 1)$ and noise scale $\sigma \sim \mathcal{U}(0, 1)$.

- The final term is a temperature sink centered at $(x_s, y_s) = (0.25, -0.5)$, with amplitude $A_{\text{sink}} < 0$ and decay $\lambda_s$.

This construction ensures that the true field is challenging test case for generalization in reconstruction models. The field looks as Figure 14.

Let us see if surrogate modeling using Chebyshev basis works for such a temperature field. If we take this ground truth and decompose it into Chebyshev basis, we get Figure 15, using the method given in Equation 47.

It can be seen that $c_{i,j}$ values above $i = 10$ and $j = 10$ are essentially zero. Hence, we consider the true coefficients for surrogate modeling to be $d = 10$ and $e = 10$.

### E.3 Reconstruction

Let us now reconstruct the field from samples taken from the temperature field. We emphasize that the approximation algorithm does not have the information of the true coefficients. We generate the learned coefficients using the training model using Chebyshev basis over the samples from the temperature field. Moreover, we consider that the samples are sparse in a region outside a certain radius from the center. Consider, it to be limitation due to instrumentation. It also makes this a problem of extrapolation or out-of-domain approximation.

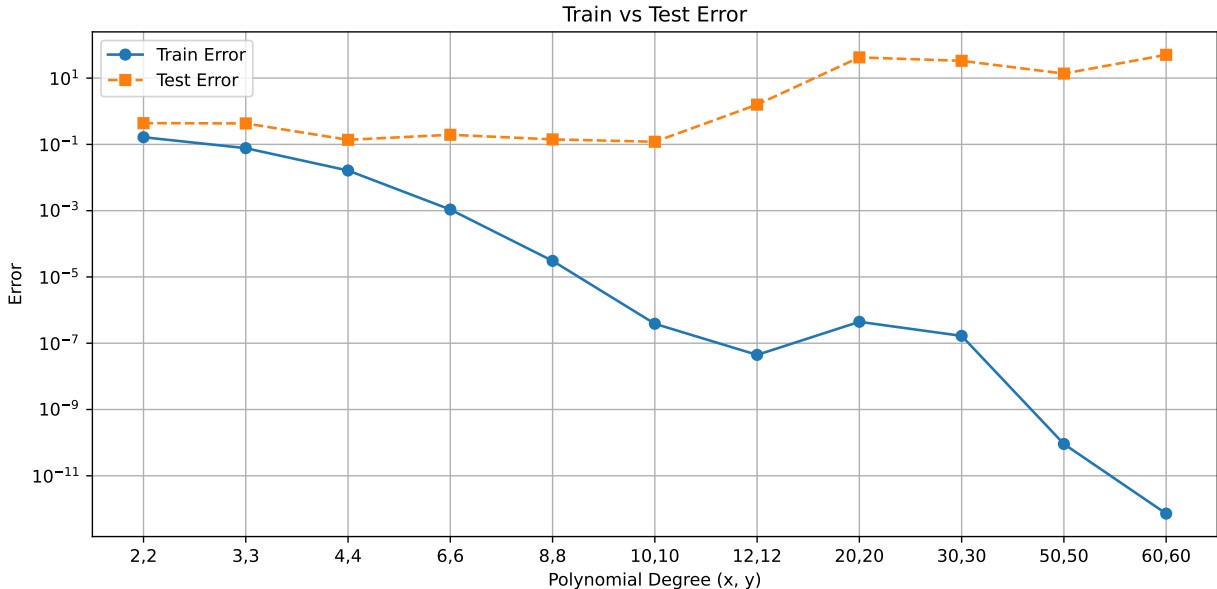

Figure 18: Train/Test error v/s order of the model basis of the surrogate for the ground truth.

We take $n_{train} = 700$ samples. We train using pseudo-inverse, as we require a closed-form solution for theoretical insights. The percentage of samples outside the radius $r = 0.75$ is 5% of the total samples. Let us see if we can reconstruct as well as extrapolate outside this radius.

We plot the reconstruction as we increase the number of parameters of the model $p$ and $q$ and compare the true coefficients to learned coefficients at $p = 10$ and $q = 10$, which is the actual effective order of the temperature field in the Chebyshev basis.

We consider the number of parameters according to the conditions of Theorem 3.1. We demonstrate the results in Figure 16

It can be seen that we have $n > d, e$ that is, we were well above the sampling threshold. The noise was sufficiently low, the surrogate approximation basis was appropriately chosen and that when the model parameters were $p, q = d, e$ we saw the best reconstruction as well as extrapolation in the sparsely sampled domain. We see that for lower model parameter values we do not have enough expressivity and for more parameter values we observe that the reconstruction degrades. This is the demonstration of the conditions given by Theorem 3.1. In fact, we also learn the exact true coefficients Figure 17.

If we just increase the order of the polynomial model. We observe that the test error is least at the order $(10, 10)$, as expected, as this is the true order of the surrogate ground truth of the true function Figure 18, or the expressive threshold of the surrogate. In short, when we approximate the true function $g(x)$ using a surrogate model $\tilde{g}(x)$, which itself is a function from a restricted function class. This introduces an approximation error due to the limited expressiveness of the surrogate itself. This adds extra error and it is very high in comparison to our observations of ideal strict generalization, as we are working with a surrogate model, where the ground truth is not exactly decomposed into a linear combination of certain basis $\Phi_t$.

This finishes our demonstration of real world application demonstration of our findings. We used a surrogate model setting to also demonstrate the limitation in which the model basis do not match exactly to the ground truth in an ideal scenario.

