# OpenReview forum: "Taxonomical Insights on Parameter Space Generalization in Over-parameterized Models"
_TMLR — Rejected by TMLR_

### Review · Reviewer_WekQ · 2025-10-20

**Summary Of Contributions:**

This work focuses on studying modern statistical phenomenon like overparametrization and double descent using the classical tools of fixed basis regression and Weierstrass approximation.  The work then goes on to describe a 'strict recovery' of the true parametrized function and a 'weak generalization' which correctly interpolates within the domain of the one-dimensional function.  Several examples are empirically explored to identify these existing phenomenon in the case of polynomial regression.  Finally, the behavior of different bases are compared across another few empirical examples, showing different interpolation threshold and generalization behavior.

Strengths:
- the terminology to emphasize strong generalization and weak generalization for polynomials
- the study of three different thresholds is a useful benefit of viewing the problem from the polynomial perspective
- the usage of several common polynomial bases allows for better insights on the generality of the behavior

**Additional Comments:**

- I do not think that "strict" and "weak" are the correct dual pairing, please consider using "strong generalization"/"weak generalization" or "strong recovery"/"weak recovery"
- In plots like Figure 3, please write the full meaning of ET and also preferably place another label next to the vertical line.  e.g. "Expressivity Threshold (p=d)"
- I recommend the use of a diverging colormap for zero-centered plots like Figure 15 and 17

**Audience:**

Yes

**Audience Explanation:**

The phenomenon of overparametrization and double descent have become staples of modern statistical theory and deserve a thorough treatment even from a perspective only investigating one-dimensional functions.  Even if insights from 1D functions do not easily generalize to high-dimensional functions, a thorough treatment of 1D polynomial generalization would give a benchmark to compare against and understand when such behavior is matching the expected low-dimensional behavior.

**Claims And Evidence:**

No

**Claims Explanation:**

This work claims to refine the understanding of generalization in over-parameterized learning in one-dimensional functions, connecting classical approximation theory with modern machine learning.  Although I think a full study on the behavior of one-dimensional models has the potential to contribute to our understanding of model overparametrization, I do not feel that the work properly supports its claim of contributing to the theory of double descent using Weierstrass approximation.  A major undertone of the writing is the implication that a double descent researcher is unfamiliar with Weierstrass's approximation theorem and that this paper somehow completely resolves any remaining confusion using the approximation theorem.  On the contrary, I think double descent researchers are very familiar with classical approximation theory as well as modern (deep network) approximation theory saying that arbitrary nice functions can be fit.  Furthermore, the question is no longer whether the model class can accurately fit the target function, but rather why does a neural network so commonly lead to a "weakly generalizing" solution.  A leading hypothesis for this is implicit bias as mentioned; however, this is given very little attention in the current work on polynomials.  The claim by the authors that convergence to the minimum norm solution has no theoretical proofs is unfounded.  These factors lead to an overall feeling of lack of engagement with the existing literature and making it difficult to support the claims of refined understanding of generalization behavior.

**Requested Changes:**

- The writing needs to emphasize a clearer connection with existing works in the field.  The Weierstrass approximation theorem simply says nothing about the actual computational learning of a polynomial approximation.  Double descent also occurs when fitting polynomials; however, the unique insights allegedly coming from Weierstrass approximation need to be better qualified.

- A discussion on the effect of basis rescaling and other algorithmic choices affecting implicit biases needs to be included.  Currently orthogonality is discussed but orthonormality is left unmentioned.  Moreover, the choice of underlying distribution (which is necessary to properly define orthogonality in a machine learning context) seems to consistently be chosen as the uniform weighting; however, this implications of this for downstream learning results seems to be not clearly mentioned.

- I think that a short exploration of the exact same nature onto a two-dimensional function would also vastly enhance the work.  Not only would the authors get to see first-hand the various choices which are possible independent of the total number of parameters but still affect the choice of parametrization (e.g. in the appendix you only considered polynomials with maximal terms of the same degree in both dimensions).  This would also show whether the expressivity threshold as you define it based on the number of parameters is valid under basic extensions.  Even the question of your definition of interpolation would be able to be more clearly defined.  For example, if you replaced Figure 16 with a nonconvex shape, is it immediately clear whether generalization would provide interpolation across the convex relaxation as interpolation or across the data support?  I think it would be valuable to understand the effects of polynomials here.

---

> ### Author Response · Authors · 2025-12-24
>
> ## Response to Requested Changes
>
> ### 1. Engagement with Double Descent and Weierstrass Theorem
>
> **Reviewer Comment:**
> *The writing implies researchers are unfamiliar with Weierstrass... the question is why models lead to a "weakly generalizing" solution. The claim that convergence to min-norm solutions has no proof is unfounded.*
>
> **Response:**
> We imply that we use bernstein basis regression merely as an example for weak generalization. It reveals that there are parameters in the model parameter space which are not same as the strict generalizing parameters and can still lead to in-domain approximation. We have revised the manuscript to acknowledge the expertise of the ML community.  We classify and rigorously define different parameters in the model parameter space in the newly added Sec. 6. We also discuss the role of **gradient descent** and initialization compared to pseudo-inverse methods in it and achievability of strict generalization in over parameterized regime. This helps us to better understand our claims which was lacking in previous version.
>
>
> ---
>
> ### 2. Basis Rescaling, Orthonormality, and Weighting
>
> **Reviewer Comment:**
> *Discuss the effect of basis rescaling and orthonormality. The choice of underlying distribution for orthogonality should be mentioned.*
>
> **Response:**
> We note that any function over the real axis can be rescaled to another domain on real axis, a standard practice when fitting with polynomials. Hence, we do not delve into the effect of rescaling perspective as polynomial regression is used as an example for strict generalization. On the choice of efffect of underlying sample distribution on the generalization we have emphasized it in Corrollary 3.2 in the Appendix. Our results of theorem 3.1 apply irrespective of sampling choice and the corrolary discusses the extra effect on choice of input distribution. We have a demonstration of randomly sampled point in **Figure 2** and **Figure 8**.
>
> - **Revision:**
>   Theorem 3.1 holds irrespective of rescaling or orthonormality choices. Rescaling is applied only for illustrative purposes with certain polynomial bases. We could also choose other non-polynomial basis and the results would apply.
>
> ---
>
> ### 3. Two-Dimensional Extensions and Convexity
>
> **Reviewer Comment:**
> *Extend analysis beyond 1D toy problems.*
>
> **Response:**
> We agree with the reviewer on 1D case, however our analysis is not restricted to toy problems but also clear mathematical distinction.  A full exploration of these effects in higher dimensional generalization and multiply input-output cases remains open. However, as the primary goal of this work is **taxonomy**, we conclude with 1D problems.
>
> ---
>
> ### 4. Terminology and Visualization
>
> **Reviewer Comment:**
> *Change "strict/weak" to "strong/weak". Use labels for ET and diverging colormaps.*
>
> **Response/Action:**
> We maintain **strict/weak generalization** as the appropriate terminology. The terms "strong" and "weak" imply a tunable parameter that converts a model from one regime to another, which it does **not**; changing from strict to weak requires modifying the model architecture itself in many cases, for example a fourier basis fixed basis regression is incapable of weak generalization. We do not explicitly use the labels for ET and IT in the plots due to lack of space on the plots. We have defined the terminology in Sec 3. namely "We call the condition where
> n = d, the “sampling threshold" (ST) and the condition where p = d, the “expressivity threshold" (ET)
> and, as already mentioned, when n = p we call it the “interpolation threshold" (IT)."
>
> Moreover, according to our refined definitions of strict generalization, we can clearly observe that strict generalization is not same as parameter recovery. They only become same in the case of identifiable models, which we agree with the reviewer was a confusion in previous version.
>
> - **Revision:**
>   Refined the definition of strict generalization to include cases where there is **no injective mapping from parameter space to function space**, hence it does not always refer to parameter recovery.

---

### Review · Reviewer_RcBi · 2025-10-29

**Summary Of Contributions:**

### Summary

This paper presents a theoretical investigation into the generalization behavior of overparameterized fixed-basis regression models, encompassing random feature models and extreme learning machines as particular instances. The authors introduce two novel conceptual thresholds, namely the sampling threshold and the expressivity threshold, which complement the established interpolation threshold and provide a refined analytical framework for understanding generalization. The work differentiates between two distinct forms of generalization: strict generalization, which requires exact recovery of the target function’s coefficients and supports out-of-domain extrapolation, and weak generalization, which ensures low test error within the training domain without recovering the underlying structure.

The authors demonstrate that strict generalization cannot be achieved in overparameterized regimes for fixed-basis regression models, thereby highlighting fundamental limitations on extrapolation. In contrast, they show that weak generalization remains attainable for in-domain tasks and substantiate this result through a theoretical argument based on the Weierstrass Approximation Theorem, establishing that weak generalization can always be realized for continuous one-dimensional functions when Bernstein bases are employed. Furthermore, the study examines the relationship between the condition number of feature matrices and model stability, offering additional insights applicable to deep neural networks and quantum machine learning.

### Strengths
1. The paper offers a clear and original conceptual framework by distinguishing between strict and weak generalization and introducing the sampling and expressivity thresholds, thereby advancing theoretical understanding of generalization in overparameterized models.
2. The analysis demonstrates strong theoretical rigor, with mathematically consistent derivations that effectively integrate classical approximation theory with contemporary machine learning principles.
3. The study provides a coherent unification of traditional basis expansion models and modern overparameterized approaches, including random feature models, neural networks, and quantum machine learning systems.

### Weaknesses
1. The analysis is limited to one-dimensional fixed-basis settings, reducing its applicability to high-dimensional or adaptive models.
2. The empirical validation is minimal and lacks quantitative evidence supporting the theoretical claims.
3. The exposition is verbose and repetitive, and clearer organization would improve readability and focus.

**Audience:**

Yes

**Audience Explanation:**

The paper addresses fundamental questions about generalization in overparameterized models, a topic of strong interest to the machine learning community. Its theoretical insights linking approximation theory with modern machine learning would appeal to researchers studying generalization and model complexity.

**Broader Impact Concerns:**

Not very applicable here.

**Claims And Evidence:**

Yes

**Claims Explanation:**

The paper’s theoretical evidence is convincing within its stated scope. The broader claims connecting the results to modern machine learning models would benefit from stronger empirical or higher-dimensional support.

**Requested Changes:**

1. The scope should be clearly defined, emphasizing that the analysis is restricted to one-dimensional continuous functions with fixed bases, and discussing how the results might generalize to higher-dimensional or adaptive settings.
2. The exposition should be streamlined by reducing repetition and improving structure, ensuring a clearer flow between definitions, theoretical results, and interpretations.
3. Additional experiments or simulations connecting the theory to modern neural networks or higher-dimensional models would strengthen the empirical support.
4. The related work section should better highlight the conceptual novelty of the proposed sampling and expressivity thresholds relative to prior studies.
5. The discussion could be expanded to provide clearer practical implications for deep learning and quantum machine learning contexts.
6. Minor formatting, typographical, and grammatical issues should be corrected to improve clarity and presentation quality.

---

> ### Author Response · Authors · 2025-12-24
> **We thank Reviewer RcBi for the thoughtful and constructive review. We are pleased that the reviewer found our conceptual framework regarding strict and weak generalization to be original and theoretically rigorous. In response to the feedback, we have significantly revised the manuscript (Version 2) to expand the empirical evidence, clarify the scope, and improve the overall flow.**
>
> ## Response to Requested Changes
>
> ### 1. Scope and High-Dimensionality
>
> **Reviewer Comment:**
> *The analysis is limited to one-dimensional fixed-basis settings; discuss how the results might generalize to higher-dimensional settings.*
>
> **Response:**
> While the foundational theorems are presented in 1D for analytical clarity, the definitions of the **sampling threshold** and **expressivity threshold** are fundamentally dimension-agnostic.
>
> - **Revision:**
>   We emphasize this in **Appendix E** through a multi-dimensional example. Specifically, we model a temperature field using bivariate Legendre polynomials. The results confirm that the **strict vs. weak generalization trade-off** and the failure of extrapolation in over-parameterized regimes trained with pseudo-inverse methods persist in multi-dimensional settings.
>
> ---
>
> ### 2. Empirical Validation
>
> **Reviewer Comment:**
> *The empirical validation is minimal and lacks quantitative evidence; additional experiments connecting theory to modern networks would strengthen support.*
>
> **Response:**
> We have added several quantitative experiments in Version 2 to substantiate our theoretical claims.
>
> - **Revision:**
>   We include a new comparison between **Moore-Penrose pseudo-inverse training** and **Gradient Descent (GD)** initialized near the true parameters (Figure 12). This showcases existence of the true solution but unachievable by pseudo inverse methods for strict genrealization case. We further discuss the convexity of the true and training loss and distinguish various parameter sets and their properties. We agree with the reviewer that the discussion on insights to deep learning is not theoretical and rather conceptual. We intend to study these different generalization behaviors in neural networks in subsequent articles.
>
> ---
>
> ### 3. Exposition and Organization
>
> **Reviewer Comment:**
> *The exposition is verbose and repetitive; clearer organization would improve readability.*
>
> **Response:**
> We performed a comprehensive editorial pass to streamline the manuscript in Version 2.
>
> - **Revision:**
>   The **Introduction** and **Conclusion** has been restructured to eliminate redundant definitions and improve clarity. We try to reduce as much verbosity as possible for the conceptual connections made in the article.
>
> ---
>
> ### 4. Conceptual Novelty and Related Work
>
> **Reviewer Comment:**
> *Highlight the conceptual novelty of the proposed sampling and expressivity thresholds relative to prior studies.*
>
> - **Revision:**
>   The **Introduction** and **Conclusion** are revised to clearly highlight the conceptual contribution and reduce any overly misinterpretable claims. The thresholds serve as **taxonomical regimes** that provide interpretability for strict generalization in the models we focus on. We agree that these thresholds are indirectly connected to several thresholds already existing in literature, as we had pointed out in the previous version, too. A proper analysis of it would be useful. However, they serve a useful purpose in understanding generalization in the models we focus on.
>
> We have added a list of changes in the current version of the article in the "Changes Since Last Submission:" section as Version 2.
>
> We believe the updated draft provides a much clearer treatment of the core theory and technical details. We look forward to your thoughts and are happy to continue polishing the work with your guidance.

---

### Review · Reviewer_rKTM · 2025-11-30

**Summary Of Contributions:**

The manuscript studies one-dimensional regression models with fixed basis functions and in-
troduces the concepts of strict generalization (perfect prediction plus exact recovery of the
“true” coefficient vector) and weak generalization (small test error only). Three thresholds
— parametrization, sampling, and expressivity — are defined in terms of the number of data
points n, the number of true basis functions d, and the number of model parameters p. The au-
thors prove that strict generalization cannot occur in the over-parameterized regime (p > n) for
fixed-basis regression, while weak generalization can still hold, drawing connections to classical
approximation theory. Numerical experiments illustrate design-matrix conditioning for various
bases.
While the topic is relevant and the narrative is coherent, I do not believe the manuscript,
in its current form, meets the novelty and depth standards expected for a research publication
in this area. Much of the analysis follows directly from elementary linear algebra, classical
approximation theory, or previously published work on double descent and benign overfitting.
The conclusions are highly dependent on restrictive modeling assumptions and the claimed
relevance to deep learning and quantum machine learning is not convincingly supported.
I therefore recommend rejection.

**Additional Comments:**

None.

**Audience:**

No

**Audience Explanation:**

See the points above. Additionally:

If the authors wish to pursue this line of work, I recommend:
1. Reframing the manuscript as an expository or tutorial piece with appropriately modest
claims.
2. Providing genuinely new quantitative results, such as risk bounds depending on (n, p, d),
noise levels, and condition numbers.
3. Extending the analysis to multi-dimensional inputs or models with learned features.
4. Clearly formalizing what is meant by “fundamental limits”: algebraic constraints vs.
minimax lower bounds.
5. Improving the clarity of definitions and reducing rhetorical claims about deep learning.

**Broader Impact Concerns:**

See my previous comments.

**Claims And Evidence:**

No

**Claims Explanation:**

1. Strict vs. weak generalization. The distinction made between strict and weak general-
ization is essentially definitional, closely paralleling the standard separation between parameter
recovery (identifiability of coefficients) and predictive risk minimization in regression. The main
impossibility result—that strict generalization cannot hold for p > n—is a direct consequence
of the fact that a fixed-basis regression model in the over-parameterized regime has a non-trivial
null space, making coefficient recovery impossible without additional structure. Theorem 3.1
merely formalizes this via necessary conditions (p ≥d, n ≥d, n ≥p, noise orthogonality, and
span inclusion), but these conditions are algebraically expected rather than fundamentally new.
Furthermore, the framework presupposes that the target function is itself given by a finite basis
expansion in the prescribed “true” basis, and that the learner’s basis contains this basis in its
span. These are strong assumptions that undermine the generality of the conclusions.
2. Limited relevance to modern over-parameterized learning. The analysis is re-
stricted to:
• one-dimensional functions on an interval;
• fixed, non-learned basis functions;
• closed-form pseudo-inverse solutions;
• arbitrary but unspecified “sampling error” vectors;
• no stochastic input distribution or optimization dynamics.
Given this very special setting, connections to deep neural networks or quantum machine learn-
ing remain speculative. The manuscript sidesteps feature learning, stochastic gradient dynamics,
implicit regularization, and noise models—all central components of contemporary theory on
benign overfitting and double descent. As a consequence, the results illuminate a specific linear
model but offer limited insight into the broader over-parameterized regime the introduction
aims to address.
3. Relation to prior work and novelty. The manuscript cites relevant literature (e.g.,
Muthukumar et al. 2020; Subramanian 2022; Peters & Schuld 2023; Poggio et al. 2019). How-
ever, it underplays the extent to which the core ideas are already present in that work:
• The interplay between sampling, model complexity, and basis choice is well known.
• Condition-number analysis of design matrices in over-parameterized linear models is not
new.
• The use of Bernstein polynomials and the Weierstrass theorem for approximation on
intervals is classical and imported verbatim from approximation theory.
Overall, the manuscript feels more pedagogical than novel: it synthesizes known facts and
illustrates them numerically, but does not deliver new theoretical insights.
4. Technical content and rigor. The main theorems follow from standard properties of the
Moore–Penrose pseudo-inverse and linear algebra. While correct, they do not represent deep
technical advances. Several issues reduce the rigor of the technical development:
• Noise is handled by requiring Φ#E = 0, a condition that is typically unattainable except
in contrived settings.
• There are no quantitative error bounds describing how predictive performance degrades
when the necessary conditions are nearly satisfied.
• The discussion of the infinite-dimensional case (d= ∞) is not reconciled with the finite-d
threshold conditions in a coherent way.
The extensive numerical section on condition numbers lacks theoretical support, providing de-
scriptive but not analytical insights.
5. Presentation and writing. The exposition is readable but imprecise at times. Terms
such as “fundamental limits,” “true basis,” and “expressivity threshold” are used in non-
standard ways. The discussion of connections to deep learning remains conceptual rather than
formal. The manuscript also contains stylistic and grammatical issues that require editing.

**Requested Changes:**

See my detailed report above.

---

> ### Author Response · Authors · 2025-12-24
> **We thank Reviewer rKTM for their rigorous critique. We acknowledge the reviewer’s perspective that some results appear algebraically expected; however, we believe the value of our work lies in formalizing a taxonomy generalization, thresholds and parameters in the model parameter space. Our value does not lie in finding metrics but in conceptually understanding.**
>
> We thank Reviewer rKTM for their rigorous critique. We acknowledge the reviewer’s perspective that some results appear algebraically expected; however, we believe the value of our work lies in formalizing a **taxonomy of generalization and parameter sets** to explain the observation of success of overparameterized models conceptually.
>
> In Version 2, we have improved our introduction and conclusion to improve the interpretation of our results.
>
> ---
>
> ## Response to Requested Changes
>
> ### 1. Novelty and “Elementary” Linear Algebra
>
> **Reviewer Comment:**
> *The impossibility result (strict generalization fails for \( p > n \)) is a direct consequence of a non-trivial null space... these conditions are algebraically expected rather than fundamentally new.*
>
> **Response:**
> We do not agree with the reviewer on this comment as strict and weak generalization is not explicitly distinguished in literature.
> While the existence of a null space is a standard result, its implication on **Generalization Taxonomy** we propose is not trivial. Most modern literature focuses on learning in strict sense and not in the weak sense as distinguished in our article. Our contribution is the formal proof that over-parameterization creates a fundamental barrier to **true feature learning** (strict generalization) if we use pseudoinverse methods, however non-existence of these solutions is not proven even if we have a null space, it can still be achievable by gradient descent method upon proper initialization. We also note that the minimum 2 norm solution provided by the pseudo-inverse methods is not the parameter vector which provides strict generalization, they only coincide in the underparameterized regime as shown by our theorem.  We have also showcased that weak generalization even if there is a null space can be achievable for in-domain approximation by training using pseudo inverse methods, through the discussion of bernstein basis.
>
> - **Revision:**
>   We have improved our definition of strict generalization and emphasize that the previous definition applied to identifiable models, which fixed-basis regression models fall into. We have added **Section 6** to clarify that the classical bias–variance trade-off derived for strict generalization applies only when pseudo-inverse solutions are used to train the model, and classical learning theory is not wrong but it only focusses on these types of generalization behavior and ignores the other type of behavior of weak one.
>
> ### 2. Relevance to Modern ML (Feature Learning & Optimization)
>
> **Reviewer Comment:**
> *The analysis sidesteps feature learning and stochastic gradient dynamics; connections to deep learning remain speculative.*
>
> **Response:**
> We agree that fixed-basis models are simpler than deep networks and do not capture full feature learning dynamics. However, the primary goal of this article is **taxonomy and conceptual clarification**, rather than modeling all aspects of modern deep learning. We therefore focus on **identifiable models**, where strict versus weak generalization can be cleanly defined and analyzed.
> Nevertheless, as rightly suggested by the reviewer we explicitly discuss how optimization dynamics changes the picture of feature learning in newly added Sec. 6.
>
> ### 3. High-Dimensionality and Generalization
>
> **Reviewer Comment:**
> *The analysis is restricted to 1D functions; extending to multi-dimensional inputs is recommended.*
>
> **Response:**
> We have addressed this limitation by moving beyond 1D toy problems in **Appendix E**, where we consider higher-dimensional input settings. We agree with the reviewer that a complete and systematic extension to general multi-dimensional feature learning remains an open question, which we explicitly acknowledge and emphasize in the **Conclusion**.
>
> ---
>
> ### 4. Rigor of Noise Models
>
> **Reviewer Comment:**
> *Noise is handled by requiring $\( \Phi^\dagger E = 0 \)$, which is unattainable in practice; there are no quantitative error bounds.*
>
> **Response:**
> This condition was not chosen arbitrarily, but follows directly from the structural requirements identified in **Theorem 3.1**. We respectfully disagree that noise effects are unaddressed: their impact is explicitly analyzed in **Section 5** and further discussed in the **Appendix**, where we examine how noise alters the "strict" generalization behavior.

---

### Comment · Action_Editor_LQ7G · 2025-12-17
**Discussion**

Hi authors and reviewers,

Authors - you haven't yet responded to the reviewer comments. The discussion period has almost ended. Please use this opportunity to respond to the reviewers.

Reviewers (especially those who have already submitted a recommendation) - I would ask that you please give the authors a little bit more time, or at least be open to changing your recommendation, if the authors do indeed respond.

---

### Decision · Action_Editor_LQ7G · 2026-01-13

**Recommendation:** Reject

**Additional Comments:**

All three reviewers recommended rejection. I think this paper would do well as a resubmission, with some tweaks to the claims being made, positioning in relation to classical analysis, and tweaks to language.

**Audience:**

Yes

**Audience Explanation:**

All three reviewers agreed that this criterion was satisfied.

**Claims And Evidence:**

No

**Claims Explanation:**

All three reviewers were indicated "no" for this question (with one of them changing their mind in the decision).

Two of the reviewers were concerned that the results in this paper "follow directly from standard linear algebra and classical approximation theory, and the conceptual distinctions largely repackage well-known ideas about parameter recovery versus prediction" and "Much of the analysis follows directly from elementary linear algebra, classical approximation theory, or previously published work on double descent and benign overfitting." A secondary and less important concern is regarding the applicability of the results, and reviewers commented on this too (one dimensional inputs, noise conditions, fixed hidden layers, etc.)

While these issues do not prevent the paper from being accurate, convincing and providing clear evidence, I think the concern of the reviewers is that the repackaging of these ideas may lead an uninformed reader to misunderstand the scope of contributions in this paper.  Also, connecting these ideas to "modern overparameterized learning, deep networks, and benign overfitting remain largely speculative and unsupported."

Some additional and more minor details were raised, and I encourage the reviewers to attend to these before resubmitting.

**Resubmission Of Major Revision:**

The authors may consider submitting a major revision at a later time.